EMBO
Molecular Medicine

# Stress signaling in breast cancer cells induces matrix components that promote chemoresistant metastasis

Jacob Insua-Rodríguez[1,2,3], Maren Pein[1,2,3], Tsunaki Hongu[1,2,4], Jasmin Meier[1,2], Arnaud Descot[1,2], Camille M Lowy[1,2,3], Etienne De Braekeleer[1,2], Hans-Peter Sinn[5], Saskia Spaich[6], Marc Sütterlin[6], Andreas Schneeweiss[7,8] & Thordur Oskarsson[1,2,9,*] 

## Abstract

Metastatic progression remains a major burden for cancer patients and is associated with eventual resistance to prevailing therapies such as chemotherapy. Here, we reveal how chemotherapy induces an extracellular matrix (ECM), wound healing, and stem cell network in cancer cells via the c-Jun N-terminal kinase (JNK) pathway, leading to reduced therapeutic efficacy. We find that elevated JNK activity in cancer cells is linked to poor clinical outcome in breast cancer patients and is critical for tumor initiation and metastasis in xenograft mouse models of breast cancer. We show that JNK signaling enhances expression of the ECM and stem cell niche components osteopontin, also called secreted phosphoprotein 1 (SPP1), and tenascin C (TNC), that promote lung metastasis. We demonstrate that both *SPP1* and *TNC* are direct targets of the c-Jun transcription factor. Exposure to multiple chemotherapies further exploits this JNK-mediated axis to confer treatment resistance. Importantly, JNK inhibition or disruption of SPP1 or TNC expression sensitizes experimental mammary tumors and metastases to chemotherapy, thus providing insights to consider for future treatment strategies against metastatic breast cancer.

**Keywords** breast cancer metastasis; chemotherapy resistance; extracellular matrix; stem cell niche; stress
**Subject Categories** Cancer; Stem Cells

## Introduction

Developing cancer cells face a number of stress-inducing insults during their lifetime and are regularly forced to evolve and adapt to changing conditions in order to propagate and remain viable. Intracellular stress can arise from overexpression of oncogenes, poor protein folding, and a high replicative state (Bartkova *et al*, 2006; Di Micco *et al*, 2006; Clarke *et al*, 2014; Minocherhomji *et al*, 2015). External factors from the microenvironment, such as suboptimal oxygen and nutrient levels or tissue tension, can also induce stress in cancer cells (Jain *et al*, 2014; Piskounova *et al*, 2015). Importantly, therapeutic intervention is likely to cause further stress to malignant cells. Thus, cancer cells that persevere despite exposure to the myriad of stress-inducing events ultimately contribute to metastatic progression and resistance to therapy.

A number of distinct internally and externally induced stress stimuli lead to the activation of c-Jun N-terminal kinase (JNK)/ stress-activated protein kinase (SAPK) family members. JNKs are mitogen-activated protein kinases (MAPKs) that include the ubiquitously expressed JNK1 and JNK2, as well as JNK3, which is specifically expressed in brain, heart, and testis (Davis, 2000). JNKs can be activated by several growth factors and cytokines and by stress-inducing signals from pathogens, radiation, and drugs, and their activity is known to influence a number of cellular functions such as differentiation, cell polarity, proliferation, and viability (Wagner & Nebreda, 2009; Hotamisligil & Davis, 2016). The *JUN* proto-oncogene encodes a key transcription factor, c-Jun, that is activated by JNK-induced phosphorylation of serines 63 and 73, and forms homo- or heterodimers with members of the FOS, ATF, and MAF protein families to constitute the transcription factor activator protein-1 (AP-1; Eferl & Wagner, 2003). Interestingly, the effects of

1 Heidelberg Institute for Stem Cell Technology and Experimental Medicine (HI-STEM gGmbH), Heidelberg, Germany
2 Division of Stem Cells and Cancer, German Cancer Research Center (DKFZ), Heidelberg, Germany
3 Faculty of Biosciences, University of Heidelberg, Heidelberg, Germany
4 Department of Physiological Chemistry and Department of Environmental Biology, Faculty of Medicine, University of Tsukuba, Tsukuba, Japan
5 Institute of Pathology, University of Heidelberg, Heidelberg, Germany
6 Department of Obstetrics and Gynecology, University Medical Centre Mannheim, Heidelberg University, Mannheim, Germany
7 National Center for Tumor Diseases—NCT, Heidelberg, Germany
8 Department of Obstetrics and Gynecology, University of Heidelberg, Heidelberg, Germany
9 German Cancer Consortium (DKTK), Heidelberg, Germany
*Corresponding author. Tel: +49 (0) 6221/42-3903; E-mail: t.oskarsson@dkfz.de

JNK pathway activation are highly context-dependent and can result in pleiotropic outcomes. This is particularly evident in cancer, where the role of the JNK pathway appears to be paradoxical. For instance, studies have shown that JNK1 and JNK2 can play tumor-suppressive and tumor-supportive roles, depending on the tumor type and molecular context (Sakurai et al, 2006; Hui et al, 2008; Das et al, 2011; Schramek et al, 2011). In addition, JNK is a recognized inducer of apoptosis under certain conditions while in other settings it promotes survival (Lin, 2003; Wagner & Nebreda, 2009). These intriguing paradoxical effects elicited by JNK activity prompted us to investigate its impact on breast cancer metastasis.

Here, we show how breast cancer cells exploit the JNK signaling pathway to promote tumor growth and metastasis. We find that JNK signaling, that is active in a subpopulation of cancer cells within tumors, induces a stem cell and wound healing gene expression program that includes a number of extracellular matrix (ECM) proteins such as osteopontin (SPP1) and tenascin C (TNC). SPP1 and TNC belong to a subgroup of ECM proteins, termed matricellular proteins, that serve as cell regulators that reside within the matrix, rather than contributing to its structure, and they are expressed in stem cell niches (Haylock & Nilsson, 2006; Chiquet-Ehrismann et al, 2014; Insua-Rodriguez & Oskarsson, 2016). We find that deficiency of either SPP1 or TNC, or inhibition of upstream JNK signaling, impairs experimental mammary tumor progression to metastasis. Moreover, chemotherapy treatment leads to JNK-mediated induction of SPP1 and TNC in metastatic breast cancer cells, indicating that, apart from its intended function as a cytotoxic agent, chemotherapy may actually reinforce the establishment of a metastatic niche via the JNK signaling pathway. Importantly, disruption of either JNK activity or expression of SPP1 or TNC sensitizes mammary tumors and metastases to chemotherapy, suggesting a potentially useful combinatorial treatment strategy to target metastatic breast cancer.

## Results

### JNK signaling in breast cancer cells promotes mammary tumor growth and lung metastasis

To investigate the functional role of JNK in metastatic breast cancer, we first analyzed JNK activity in clinical effusion samples (ascites and pleural effusions) from 10 breast cancer patients with metastatic disease. Western blot analysis revealed that 10/10 samples exhibited high expression of active JNK, and 9/10 samples showed high levels of the activated downstream JNK-induced transcription factor c-Jun, when compared to normal human mammary epithelial cells (Appendix Fig S1A–C). To determine a potential connection between JNK activity and patient outcome, we analyzed tissue microarrays of tumor samples from breast cancer patients with recurring disease and found that high JNK activity was associated with poor overall survival (Fig 1A; Appendix Fig S1D and Appendix Tables S1 and S2). Moreover, immunohistochemical analysis of matched patient-derived primary breast tumors and lung metastases revealed heterogeneous JNK signaling within both primary tumors and metastases, based on activated c-Jun (Fig 1B). A significant increase in the number of cells with

active c-Jun was observed in lung metastases compared to their respective primary tumors (Fig 1C). This suggested a link between JNK signaling and poor outcome in breast cancer and therefore prompted us to analyze the JNK pathway during metastatic progression. We used two human metastatic breast cancer cell lines, MDA231-LM2 and SUM159-LM1, that exhibit reproducible metastasis to the lungs (Minn et al, 2005 and Appendix Fig S2). Indeed, both cell lines showed high JNK activity (Appendix Fig S3A). Moreover, immunofluorescence analysis of mammary tumors and matched metastatic nodules in lungs from mice injected with MDA231-LM2 cells revealed cancer cell-specific intracellular c-Jun activity (Appendix Fig S3B and C). In line with our results from human breast cancer patients, the proportion of cancer cells expressing active c-Jun was significantly higher in lung metastases when compared to growing, matched mammary tumors (Fig 1D). Furthermore, analysis of cancer cells with active c-Jun in metastatic nodules of lungs from mice injected with MDA231-LM2 breast cancer cells revealed that over 50% of cancer cells in micrometastases (day 7 post-injection) exhibited active JNK signaling whereas less than 15% were observed in macrometastases (day 21 post-injection), often associated with the invasive front (Fig 1E and F). This suggested that cancer cells in breast tumors and metastases display heterogeneity with respect to JNK activity and established a link between JNK signaling and metastatic progression.

To address experimentally whether JNK signaling contributes to breast cancer metastasis to lungs, MDA231-LM2 or SUM159-LM1 cells were injected bilaterally into the fourth mammary fat pads of NOD.Cg-$Prkdc^{scid}$ $Il2rgtm1^{Wjl}$/SzJ (NSG) mice. Since JNK activity could be efficiently repressed by the second-generation, ATP-competitive, anthrapyrazolone JNK inhibitor CC-401 (hereafter referred to as JNKi) in MDA231-LM2 and SUM159-LM1 cells (Fig EV1A), tumor-bearing mice were treated with JNKi every third day, starting at day 5 post-implantation (Fig 1G). The mice did not display overt symptoms of toxicity or loss of weight. However, treatment with JNKi caused a significant reduction in both mammary tumor growth and pulmonary metastasis in MDA231-LM2 and SUM159-LM1 xenograft models, as assessed by tumor volume and number of human vimentin-positive foci in lung sections (Fig 1H–J). To analyze JNK function during metastatic colonization from circulation, luciferase-transduced MDA231-LM2 cells were injected intravenously into NSG mice and the mice treated concurrently with JNKi (Fig 1K). The treatment was continued every third day, and metastatic colonization was analyzed by bioluminescence imaging at day 17. Mice treated with JNKi exhibited significantly reduced metastatic burden in the lungs as compared to vehicle-treated controls, indicating that active JNK promotes metastatic colonization (Fig 1L and M).

To determine whether endogenous JNK signaling in cancer cells is required during metastatic colonization, we intravenously injected MDA231-LM2 cells that were pre-treated in vitro with JNKi. In culture, these cells exhibited no differences in the rate of cell growth or levels of apoptosis when compared to untreated cells (Fig EV1B and C). However, following intravenous injection, JNKi pre-treatment caused a marked reduction in lung colonization ability as determined by bioluminescence (Fig EV1D–F). Together, these results suggest that endogenous JNK signaling in cancer cells is critical for breast cancer progression and lung metastasis.

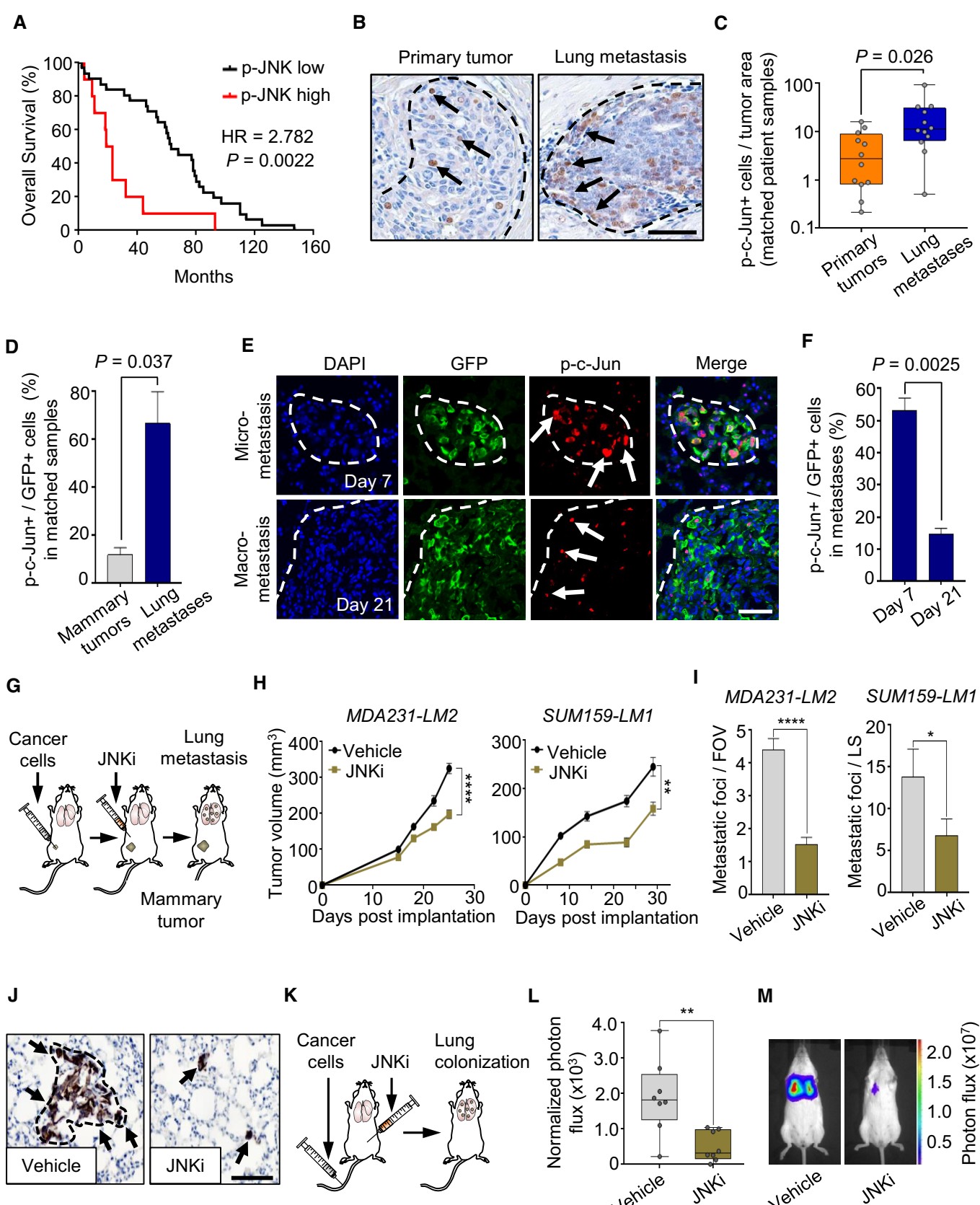

**Figure 1.**

◀

**Figure 1.   JNK signaling mediates mammary tumor growth and metastasis to lungs.**

A    Kaplan–Meier analysis of overall survival in breast cancer patients with recurrent lethal disease, classified according to p-JNK expression. p-JNK was determined by immunostaining of breast cancer samples in a tissue microarray; p-JNK low, *n* = 31; p-JNK high, *n* = 10. *P*-value and hazard ratio (HR) were calculated by log-rank test.

B    Immunostaining of p-c-Jun (arrows) in patient-matched breast tumor and lung metastasis. Scale bar, 50 μm.

C    Quantification of p-c-Jun-positive (p-c-Jun+) cells per tumor area in matched primary tumors and lung metastases. Boxes depict median from 12 patients with upper and lower quartiles. Whiskers show maximum and minimum. *P*-value was determined by two-tailed paired Student's *t*-test.

D    Quantification of p-c-Jun-positive (p-c-Jun+) cancer cells within mammary tumors and matched metastases in mice injected with MDA231-LM2 cells. Values are mean from three mice per group ± SEM. *P*-value was determined by a two-tailed paired Student's *t*-test.

E    Immunofluorescence analysis of p-c-Jun in micrometastatic (day 7) and macrometastatic (day 21) nodules in a xenograft mouse model injected intravenously with MDA231-LM2 cells. Arrows indicate p-c-Jun expressing cancer cells. DAPI was used to stain cell nuclei, and cancer cells express GFP as a marker. Scale bar, 50 μm.

F    Quantification of p-c-Jun+ cancer cells in panel (E) showing mean from three mice per group ± SEM. *P*-value was determined by two-tailed Mann–Whitney test.

G    Experimental setup to address the role of JNK signaling in mammary tumor progression and metastasis. MDA231-LM2 or SUM159-LM1 cells were implanted bilaterally into the fourth mammary fat pads of NSG mice. *In vivo* treatment with the JNK inhibitor CC-401 (JNKi) commenced at day 5 post-implantation and was repeated every 3 days thereafter until analysis.

H    Mammary tumor growth curves from mice described in panel (G). Each value represents the mean ± SEM. For MDA231-LM2, *n* = 12 tumors in six mice for each experimental group. For SUM159-LM1, *n* = 16 tumors in eight mice for each experimental group.

I, J    Lung metastatic burden in MDA231-LM2 or SUM159-LM1 tumor-bearing mice treated with JNKi. (I) Quantification of lung metastases. FOV, field of view; LS, lung section. Each value represents the mean ± SEM. For metastatic foci per FOV, 8–10 fields were quantified per mouse. MDA231-LM2 tumor-bearing mice: *n* = 6 mice per group. SUM159-LM1 tumor-bearing mice, vehicle *n* = 8 mice; JNKi *n* = 7 mice. (J) Representative histology examples of metastasis in lungs, detected by expression of human vimentin. Scale bar, 100 μm.

K    Experimental setup to determine lung colonization of *in vivo* vehicle-treated and JNK-inhibited MDA231-LM2 cancer cells after intravenous injection. *In vivo* treatment with JNKi commenced at the time of implantation and was repeated every 3 days thereafter until analysis at day 17.

L    Metastatic burden determined by photon flux in lungs of MDA231-LM2-injected NSG mice treated with vehicle and JNKi. Boxes show the median with upper and lower quartiles. Whiskers represent minimum and maximum values. Vehicle control, *n* = 8; JNKi-treated, *n* = 8.

M    Examples of bioluminescence in mice that were quantified in panel (L).

Data information: For panels (H, I, and L), *\*P* < 0.05, *\*\*P* < 0.01, *\*\*\*\*P* < 0.0001. *P*-values were determined by a two-tailed Mann–Whitney test.

## JNK signaling induces expression of genes associated with ECM, wound healing, and stem cell properties

To identify molecular mediators of JNK-induced metastasis, we employed loss- and gain-of-function approaches to modulate JNK activity in breast cancer cells. To reduce JNK signaling, we exposed MDA231-LM2 cells to JNKi for 48 h. To increase JNK activity, we ectopically expressed a constitutively active form of JNK, consisting of a protein fusion between JNK1 and its upstream MAPK kinase (MAPKK) activator MKK7 (MKK7-JNK), or a mutated version (MKK7-JNK(mut)) in which the phosphorylation motif Thr180-Pro-Tyr182 in JNK1 is replaced with Ala-Pro-Phe, thereby preventing its activation by MKK7 (Derijard *et al*, 1994; Lei *et al*, 2002). Addressed by Western analysis, expression of MKK7-JNK, but not MKK7-JNK (mut), in MDA231-LM2 cells led to a robust induction of the JNK pathway (Appendix Fig S4A). To determine the cellular processes affected by JNK signaling in breast cancer cells, global gene expression changes induced by these loss- and gain-of-function approaches were analyzed in parallel to identify genes that were most sensitive to modulation of JNK activity (Fig 2A) and by gene ontology (GO) using Database for Annotation, Visualization and Integrated Discovery (DAVID). This analysis revealed that JNK signaling induces genes involved in wound healing, organ development, motility, and components of the ECM (Fig 2B and Appendix Fig S4B). The association with GO terms such as wound healing and organ development suggested a possible connection to stem cell properties. We therefore explored this putative link by gene set enrichment analysis (GSEA) of a number of gene sets characterizing stem cells of the mammary gland. Indeed, genes expressed in mammary gland stem cells (Pece *et al*, 2010) were enriched in cells overexpressing MKK7-JNK and underrepresented in cells treated with JNKi (Fig 2C and D). The association between JNK activity and mammary stem cell gene signatures was consistent when we performed GSEA using four other distinct gene sets from

**Figure 2.   JNK signature in breast cancer cells is linked to ECM, stem cell, and wound healing gene networks and is enriched in basal-like breast cancer.**    ▶

A    Scatter plot elucidating fold change (FC) in gene expression between samples treated with JNKi compared to vehicle (*y*-axis) and expression of constitutively active JNK (MKK7-JNK) versus vector control (*x*-axis) from biological triplicates. Representative examples of genes that are specifically induced by JNK in metastatic breast cancer cells are highlighted in red.

B    Biological functions regulated by the JNK pathway in MDA231-LM2 cells determined by gene ontology using DAVID analysis. False discovery rate (FDR) < 0.1. BP, biological process; CC, cellular compartment; BH, Benjamini–Hochberg.

C, D    GSEA graphs showing (C) enrichment of a mammary stem cell signature (Pece *et al*, 2010) in MDA231-LM2 cancer cells expressing MKK7-JNK and (D) underrepresentation of the same mammary stem cell signature in cancer cells treated with JNKi. NES, normalized enrichment score. *P*-values were determined by random permutation test.

E    GSEA showing significant enrichment of four distinct signatures of mammary stem cells within data sets from MDA231-LM2 cells either expressing MKK7-JNK or treated with JNKi. Mammary stem cell signatures are indicated as follows: a, up in mammary stem cells (Lim *et al*, 2010); b, up in fetal mammary stem cells (Spike *et al*, 2012); c, up in adult mammary stem cells (Spike *et al*, 2012); and d, up in mammary repopulating units (Stingl *et al*, 2006). *\*P* < 0.05, *\*\*P* < 0.01, *\*\*\*P* < 0.001. *P*-values were determined by random permutation test.

F    Heat map of the JNK signature composed of 68 genes that are induced by expression of MKK7-JNK and repressed by JNKi, as analyzed in panel (A).

G    Proportion of JNK signature-positive tumors in breast cancer patients from the METABRIC data set classified according to PAM50 intrinsic biological subtypes (Chia *et al*, 2012; Curtis *et al*, 2012). *\*\*\*\*P* < 0.0001. JNK-S, JNK signature as in panel (F). Subtype patient numbers: Luminal, *n* = 1,213; HER2, *n* = 240; basal-like, *n* = 331. *P*-value was determined by Fisher's exact test.

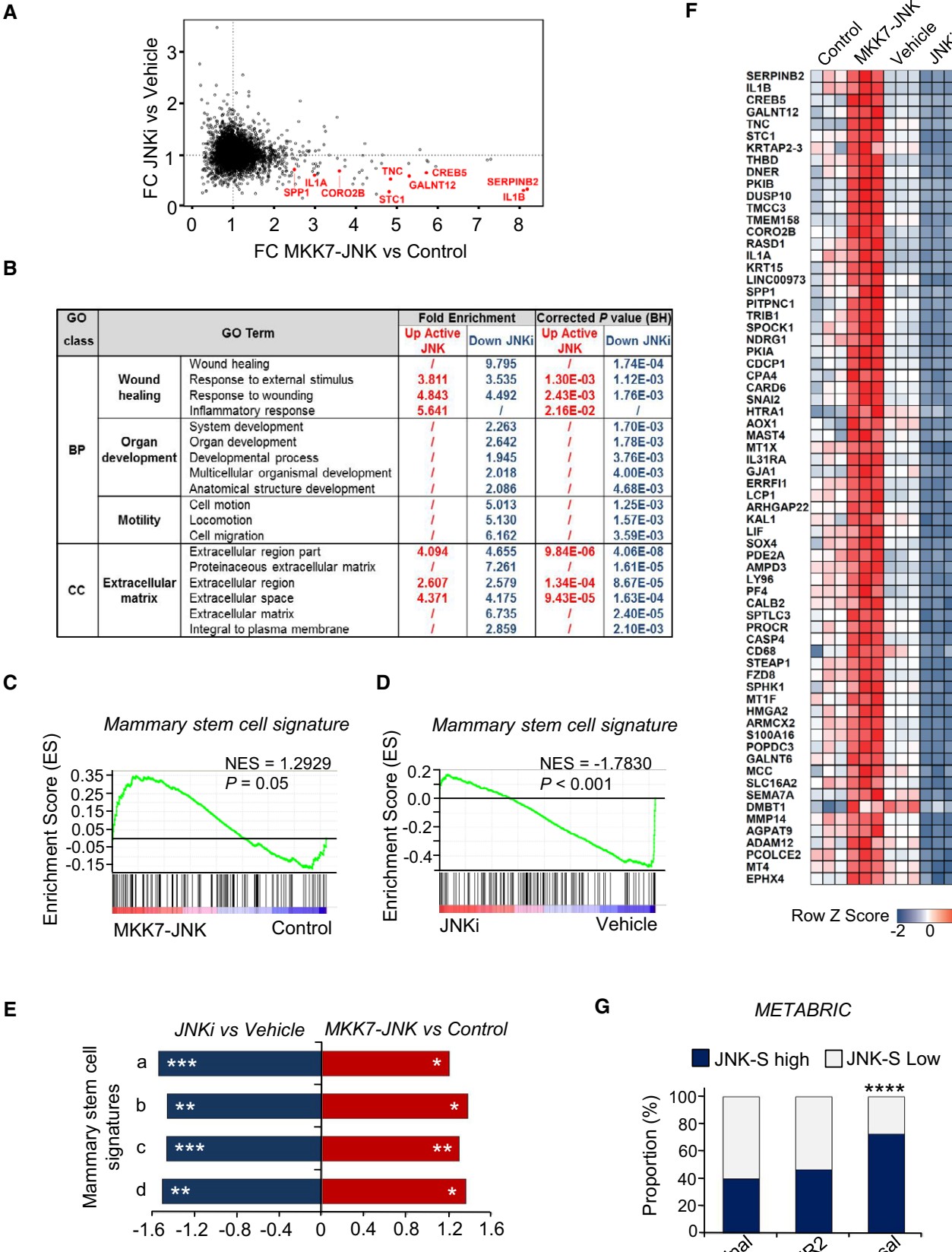

**Figure 2.**

three different studies (Stingl *et al*, 2006; Lim *et al*, 2010; Spike *et al*, 2012; Fig 2E). This suggests that, in addition to a wound healing response, JNK signaling promotes the induction of mammary stem cell properties in breast cancer cells.

Breast cancer can be divided into a number of distinct subtypes, some of which display stem/progenitor cell attributes. To analyze the relationship between JNK activity and breast cancer subtypes, we generated a JNK response signature composed of 68 genes that were both significantly induced by active JNK and significantly repressed by treatment with JNKi (Fig 2A and F). We applied the signature to breast cancer samples from the METABRIC data set (Curtis *et al*, 2012) to address the potential association between JNK signaling and breast cancer subtypes. Patient samples were classified according to JNK activity using the JNK-induced gene signature (JNK-S), and the proportion of breast cancer subtypes was analyzed according to the PAM-50 breast cancer intrinsic classifier (Chia *et al*, 2012). All subtypes included cases with JNK-S high tumors; however, a significant enrichment was observed in basal-like breast cancer compared to luminal- or HER2-positive breast cancer samples (Fig 2G). Since basal-like breast cancer is highly aggressive and is known to exhibit stem/progenitor cell characteristics (Rakha *et al*, 2008), these data further support a model in which JNK signaling promotes the acquisition of mammary stem cell attributes in breast tumors.

### JNK activity promotes stem cell properties and tumor-initiating capacity in breast cancer cells

To further investigate the association between JNK signaling and stem cell properties, we utilized a sphere culture assay that is recognized to enrich for stem cells from multiple tissues (Doetsch *et al*, 1999; Dontu *et al*, 2003; Huang *et al*, 2015). MDA231-LM2 cells were placed on ultra-low-adhesive plates in the absence of serum to form oncospheres. Gene expression profiling of these oncospheres, compared to the same cells cultured as a monolayer, revealed upregulation of several genes from the JNK signature (Fig 3A). This was validated by GSEA (Fig 3B), and Western analysis of oncospheres after 7 and 14 days in culture confirmed activation of JNK (Fig 3C). Interestingly, we also observed that gene classifiers for basal-like breast cancer, basal cells of the mammary gland and mammary stem cells, were all significantly enriched in the oncospheres (Fig 3D and E, Appendix Fig S5).

To determine whether JNK activity is required for oncosphere formation, we generated spheres using SUM159 parental cells overexpressing either MKK7-JNK or MKK7-JNK(mut). Expression of MKK7-JNK, but not MKK7-JNK(mut), significantly increased sphere formation (Fig 3F). Consistently, treatment of SUM159-LM1 cells or cancer cells from breast cancer patients with JNKi significantly impaired their sphere-forming ability (Fig 3G–J). This indicates that JNK signaling is required to maintain stem cell traits in breast cancer cells.

We next questioned whether JNK activity promotes tumor initiation *in vivo* by injecting limiting numbers of JNKi-treated MDA231-LM2 cells subcutaneously into NSG mice (Fig 3K). Indeed, JNK inhibition reduced the tumor-initiating ability of the cancer cells (Fig 3L and Appendix Fig S6). Since our transcriptomic analysis revealed a link between JNK signaling and GO terms associated with wound healing and cell motility (Fig 2B), we sought to understand the functional role of JNK in regulating the invasive and migratory abilities of breast cancer cells. To this end, we analyzed invasion of MDA231-LM2 and SUM159-LM1 cells through Matrigel after treatment with JNKi and found that both cell lines showed a significant reduction in invasiveness upon JNK inhibition (Figs 3M and EV2A). Using an *in vitro* wound healing assay, we found that JNK inhibition significantly reduced the migratory abilities of MDA231-LM2 and SUM159-LM1 cells (Fig EV2B and C). Furthermore, ectopic expression of MKK7-JNK, but not MKK7-JNK(mut), increased migration of the cancer cells (Fig EV2D and E). These results suggest that JNK activity promotes invasion and migration of breast cancer cells.

### SPP1 and TNC are regulated by JNK signaling and promote the development of lung metastasis

Two of the most significantly changed genes within the JNK signature were the matricellular proteins SPP1 and TNC that are glycoproteins within the ECM. SPP1 and TNC have been shown to be expressed in adult stem cell niches, playing a role in stem/progenitor cell maintenance and are highly induced during wound healing where they promote cell viability and migration (Haylock & Nilsson, 2006; Chiquet-Ehrismann *et al*, 2014; Insua-Rodriguez & Oskarsson, 2016). To validate the regulation of *SPP1* and *TNC* by JNK signaling, we treated MDA231-LM2 and SUM159-LM1 cells with increasing concentrations of JNKi and analyzed *SPP1* and *TNC* mRNA levels. Cells treated with JNKi showed a dose-dependent reduction in *SPP1* and *TNC* expression (Figs 4A and EV3A). Furthermore, overexpression of MKK7-JNK, but not MKK7-JNK(mut) induced high levels of both *SPP1* and *TNC* expression in MDA231-LM2 and SUM159-LM1 cells (Figs 4B, and EV3B and C). Consistently, we observed a considerable downregulation of *SPP1* and *TNC* levels in breast cancer cells from primary patient-derived effusion samples treated with JNKi (Fig 4C). Since JNK activity was induced in cell line-derived oncospheres (Fig 3A–C), we analyzed gene expression in patient-derived cells grown as sphere cultures and found a significant induction of *SPP1* and *TNC* levels (Fig EV3D). Inhibition of JNK abrogated sphere culture-induced expression of *SPP1* and *TNC* (Fig 4D). To determine whether *SPP1* and *TNC* are directly induced by the transcription factor c-Jun downstream of JNK, we performed chromatin immunoprecipitation (ChIP), followed by qPCR using two PCR primer pairs designed to flank predicted c-Jun binding sites within the promoter of each gene (Fig 4E). ChIP-qPCR analysis showed an enrichment of both *SPP1* and *TNC* promoter fragments in SUM159-LM1 and MDA231-LM2 cells when chromatin was pulled down using an antibody against c-Jun (Figs 4F and EV3E). To determine whether the link between JNK signaling and expression of *SPP1* and *TNC* is clinically relevant, we analyzed transcriptomic data sets from dissected, patient-derived, metastatic nodules (Zhang *et al*, 2009) for expression of *SPP1* and *TNC*. Indeed, metastasis samples showed a significant correlation between *SPP1* and *TNC* expression and between expression of each of the genes and *JUN* which encodes the transcription factor c-Jun (Figs 4G and EV4A). Moreover, when we analyzed *SPP1* and *TNC* expression in human breast cancer metastasis samples ranked according to JNK signature score, a marked increase in expression of *SPP1* and *TNC* was observed in metastasis samples with high JNK activity (Fig EV4B). These results suggest that JNK signaling promotes the expression of *SPP1* and *TNC* in breast cancer cells via c-Jun to promote metastasis.

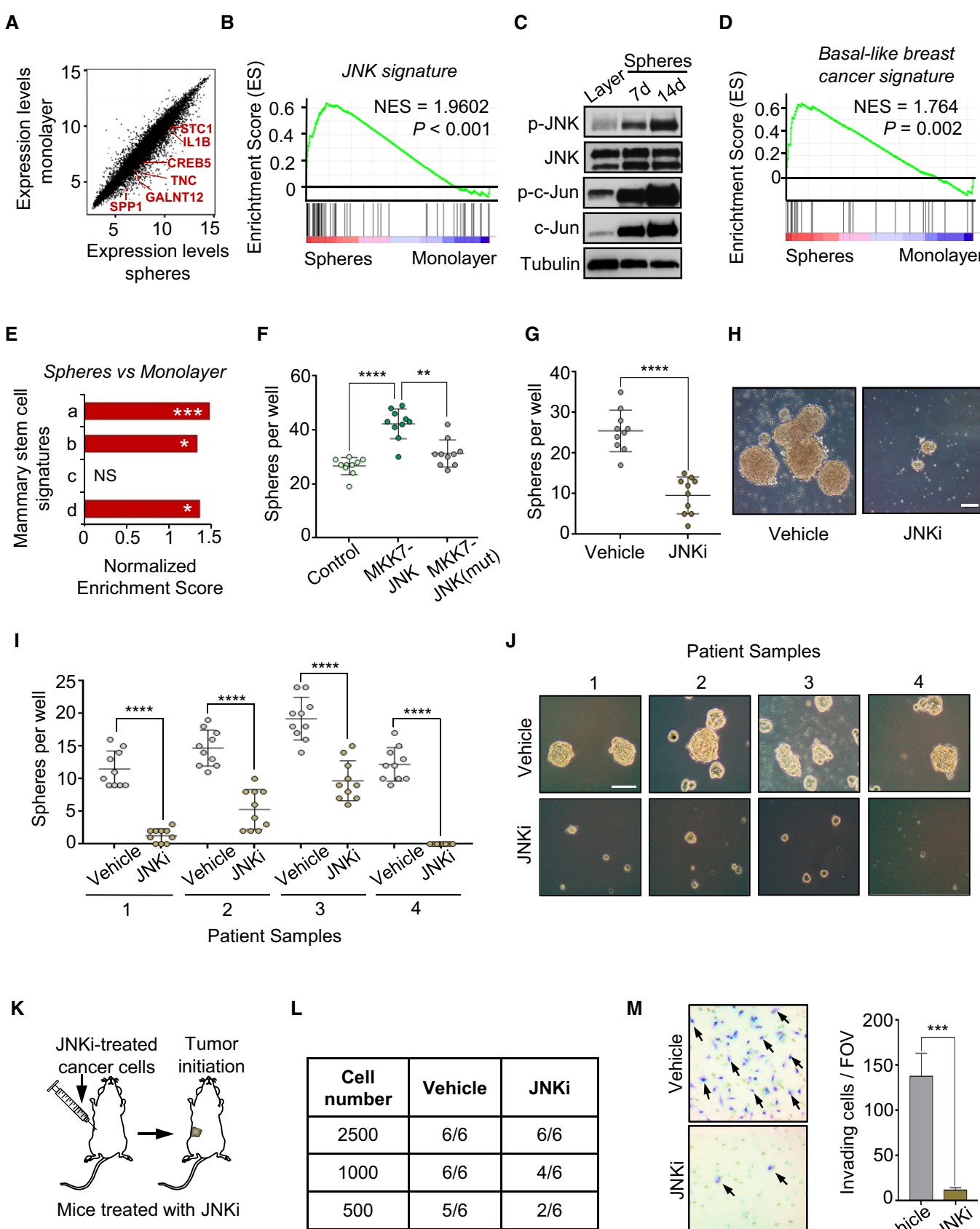

**Figure 3.**

**Figure 3. JNK signaling promotes stem cell properties and invasiveness in breast cancer cells.**

A Gene expression profile of MDA231-LM2 cells grown as oncospheres or as a monolayer. Examples of genes that are both upregulated in spheres and induced by JNK signaling are highlighted in red.

B GSEA graph showing enrichment of the JNK signature in MDA231-LM2 cancer cells grown as oncospheres. NES, normalized enrichment score. *P*-values were determined by random permutation test.

C Immunoblot of phosphorylated JNK (p-JNK), JNK, phosphorylated c-Jun (p-c-Jun), and c-Jun in oncospheres from SUM159-LM1 cells. Tubulin was included as a loading control.

D GSEA of a basal-like breast cancer signature (Perou *et al*, 2000) induced in MDA231-LM2 oncospheres. NES, normalized enrichment score. *P*-values were determined by random permutation test.

E Normalized enrichment scores derived by GSEA of mammary stem cell signatures in oncospheres from MDA231-LM2 cells. Signatures a-d are the same as described in Fig 2E. *P < 0.05, ***P < 0.001. NS, not significant. *P*-values were determined by random permutation test.

F Generation of oncospheres in SUM159 cells expressing constitutively active JNK or inactive mutated JNK. Values are mean from 10 wells per group ± SD.

G Oncosphere formation in SUM159-LM1 cells following treatment with JNKi or vehicle control. Values are mean from 10 wells per condition ± SD.

H Representative images of spheres from panel (G). Scale bar, 100 μm.

I, J Oncosphere formation in JNKi-treated cancer cells from breast cancer patients (pleural effusions or ascites). (I) Quantification of oncospheres per well in four different patients. Values are mean from 10 wells ± SD. (J) Representative oncospheres from patient samples. Scale bar, 50 μm.

K Subcutaneous injection of limiting numbers of JNKi-treated MDA231-LM2 cells in NSG mice.

L Analysis of tumor initiation ability of vehicle- and JNKi-treated cells from experiment described in panel (K); n = 3 mice per group, two implantations per mouse.

M Matrigel invasion assay of MDA231-LM2 cells treated with JNKi. Left, examples of invading cancer cells stained with crystal violet (arrows). Scale bar, 50 μm. Right, quantification of invading cells per field of view (FOV) showing mean from 8 fields ± SEM.

Data information: For panels (F, G, I, and M), **P < 0.01; ***P < 0.001; ****P < 0.0001. *P*-values were determined by a two-tailed Mann–Whitney test.
Source data are available online for this figure.

To determine whether SPP1 furthers metastatic progression in the lungs, we first performed shRNA-mediated knockdown of *SPP1* expression in MDA231-LM2 cells. Knockdown efficiency was between 80 and 90% (Appendix Fig S7A). We observed *Spp1* expression in the stroma of growing metastases. Therefore, to clear the experimental setting from any source of SPP1, the *SPP1*-knockdown cells were injected into mammary fat pads of *Spp1*$^{-/-}$ mice (Liaw *et al*, 1998) that contained the immunocompromising mutations *Prkdc*$^{scid}$ *Il2rg*$^{tm1Wjl}$ (*scid Il2rg*$^{null}$) on a mixed genetic background. *SPP1* deficiency did not affect mammary tumor growth in this model (Appendix Fig S7B). However, metastatic colonization was significantly reduced as determined by *ex vivo* lung bioluminescence and histology (Fig 4H and I). We next targeted TNC expression in SUM159-LM1 cells by shRNA-mediated knockdown (Appendix Fig S7C), injected the cells intravenously into NSG mice, and found that metastatic colonization of the lungs was also significantly reduced by TNC deficiency (Fig 4J and K). Considering this, we evaluated the clinical relevance of *SPP1* and *TNC* expression in tumors, by analyzing transcriptomic profiles from breast cancer patients that were annotated for lung metastasis (Minn *et al*, 2005, 2007). Patients with tumors expressing high levels of *SPP1* and *TNC* were associated with significantly worse lung metastasis-free survival compared to *SPP1*- and *TNC*-low patients (Fig 4L). These results indicate that expression of *SPP1* and *TNC* promotes

**Figure 4. *SPP1* and *TNC* are induced by active JNK and promote lung metastasis.**

A *SPP1* and *TNC* expression in the indicated breast cancer cells treated with increasing concentrations of JNKi. Values represent the mean of qPCR triplicates ± SD.

B *SPP1* and *TNC* expression in MDA231-LM2 cells transduced with pLVX-puro lentivirus without a transgene (control) or expressing constitutively active JNK (MKK7-JNK) or mutated inactive JNK (MKK7-JNK(mut)). Values represent the mean of qPCR triplicates ± SD.

C *SPP1* and *TNC* expression in pleural effusion (1 and 2) and ascites (3 and 4) samples from four different patients treated with JNKi (4 μM, 48 h) in culture. Values are mean ± SD from triplicates. The results from all four patient samples were included to calculate significance of JNKi on gene expression. *SPP1*, P = 0.0079; *TNC*, P = 0.0002. *P*-values were determined by two-tailed paired Student's t-test.

D Analysis of *SPP1* and *TNC* mRNA levels in SUM159-LM1 cells grown as monolayer or spheres and treated with vehicle or JNKi. Values represent the mean of qPCR triplicates ± SD.

E, F ChIP analysis of c-Jun binding to sequences in *SPP1* or *TNC* promoters. (E) Diagrams of *SPP1* and *TNC* promoters showing positions of primer pairs, flanking c-Jun/AP-1 consensus sites, that were used for qPCR following ChIP with an antibody against c-Jun. (F) Analysis of *SPP1* and *TNC* promoter fragments pulled down by c-Jun and IgG antibodies. Values represent the mean of qPCR triplicates ± SD.

G Correlation analysis of *SPP1* and *TNC* with *JUN* expression in dissected metastatic nodules from breast cancer patients (GSE14020). N = 65 metastasis samples. *P*-values were acquired by two-tailed Student's t-test.

H, I Lung metastasis in *Spp1*$^{+/-}$ mice implanted with MDA231-LM2 cells expressing control shRNA (control), and *Spp1*$^{-/-}$ mice implanted with MDA231-LM2 cells independently expressing one of two different *SPP1* shRNAs (*SPP1*-deficient). *Ex vivo* lung luminescence was analyzed, panel (H). Boxes show the median value with upper and lower quartiles. Whiskers represent minimum and maximum values. Control n = 16 mice, SPP1 deficient (1) n = 14 mice, and SPP1 deficient (2) n = 15 mice. *P*-values were determined by a two-tailed Mann–Whitney test. (I) Representative histological images of metastatic foci (top) and *ex vivo* lung bioluminescence (bottom). In histological samples, immunohistochemistry was used to determine metastatic nodules based on expression of human vimentin (arrows). Scale bar, 100 μm.

J, K SUM159-LM1 cells expressing either control shRNA (control), or one of two different *TNC* shRNAs (shTNC 1 and 2), were injected intravenously into NSG mice. (J) Metastatic colonization was quantified by lung bioluminescence. Boxes show the median value with upper and lower quartiles. Whiskers represent minimum and maximum values. Control and shTNC (1) n = 8 mice per group, shTNC (2) n = 7 mice. *P*-value was determined by a two-tailed Mann–Whitney test. (K) Representative images of metastatic foci (top) and *ex vivo* lung bioluminescence (bottom). Arrows denote vimentin-expressing cancer cells in metastases. Scale bar, 100 μm.

L Kaplan–Meier analysis of breast cancer patients (combined GSE2603 and GSE5327 data sets) showing a link between high *SPP1* and *TNC* expression and poor lung metastasis-free survival. Patient samples, *SPP1* and *TNC* low n = 70; *SPP1* and *TNC* high n = 70. HR, Hazard ratio. *P*-value was determined by log-rank test.

 

**Figure 4.**

metastatic colonization of the lungs in experimental mouse models for breast cancer and is associated with lung metastasis in breast cancer patients.

## Chemotherapy treatment induces *SPP1* and *TNC* expression in breast cancer cells via JNK signaling

Chemotherapy can deliver high levels of cytotoxic stress to cancer cells and has been shown to activate intracellular stress signaling pathways (Herr & Debatin, 2001). To investigate gene responses of metastatic breast cancer cells to chemotherapy and association with JNK signaling, we exposed MDA231-LM2 cells to the commonly used chemotherapeutic agent paclitaxel and performed transcriptomic analysis using microarrays (Fig 5A). Interestingly, close to 40% of the genes induced by paclitaxel were also induced in sphere cultures (Fig 5B), and paclitaxel-treated MDA231-LM2 cells showed a significant enrichment for both mammary stem cell signatures and activation of JNK signaling (Figs 5C, and EV5A and B). Notably, immunofluorescence analysis of macrometastases in lungs of mice injected intravenously with MDA231-LM2 and treated with paclitaxel at week 3 post-injection showed a striking induction of JNK signaling in cancer cells, as determined by activated c-Jun (Fig 5D and E). The proportion of cancer cells expressing active c-Jun was below 20% in vehicle-treated metastases and increased to over 80% in paclitaxel-treated metastases (Fig 5E). We performed Western blot analysis to confirm that treatment with paclitaxel as well as with doxorubicin, another chemotherapeutic agent that is commonly used to treat metastatic breast cancer, indeed robustly induced JNK signaling in breast cancer cells (Figs 5F and EV5C). We next questioned whether the JNK-regulated downstream targets, *SPP1* and *TNC*, were also induced in MDA231-LM2 and SUM159-LM1 cells treated with these and other relevant chemotherapies. Analysis by qPCR revealed upregulation of both genes in response to paclitaxel, doxorubicin, 5-fluorouracil, and methotrexate (Fig 5G–I; Appendix Fig S8). Notably, induction of *SPP1* and *TNC* in response to paclitaxel was blunted upon inhibition of JNK activity (Fig 5J). These results suggest that exposure to chemotherapy induces expression of the niche components SPP1 and TNC in breast cancer cells via the JNK signaling pathway.

## Disruption of *SPP1* and *TNC* expression sensitizes growing mammary tumors and metastases to chemotherapy

To determine whether SPP1 and TNC promote metastatic progression following exposure to chemotherapy, we first implanted *SPP1*-knockdown MDA231-LM2 and SUM159-LM1 cells (Appendix Figs S7A and S9A) into *Spp1*$^{-/-}$ *scid Il2rg*$^{null}$ mice. Control and shSPP1-transduced cancer cells were injected bilaterally into the fourth mammary fat pads of the animals. At 10 days post-implantation, mice were treated with paclitaxel and received a new dose every 5 days until analysis (Fig 6A). *SPP1* deficiency resulted in a significant reduction of mammary tumor growth and lung metastasis when treated with paclitaxel (Fig 6B and C) as well as combination treatment with doxorubicin (also known as adriamycin) and cyclophosphamide (AC regimen, Fig 6D–F; Appendix Fig S9B). The striking reduction in lung metastasis may be partially explained by decreased mammary tumor growth. However, neither *SPP1* deficiency nor chemotherapy alone resulted in reduced mammary tumor growth, yet chemotherapy or *SPP1* deficiency individually caused a moderate and a significant reduction in lung metastasis, respectively. These findings suggest that SPP1 is an important mediator of chemotherapy resistance.

To address whether TNC also promotes resistance to chemotherapy, we implanted *TNC*-knockdown MDA231-LM2 cells into the mammary fat pads of NSG mice and initiated treatment with paclitaxel at 1 week post-implantation (Fig 6G and Appendix Fig S10). This treatment schedule was used to minimize the potential for *Tnc* induction in the stroma of growing mammary tumors and metastases, as previously observed (Oskarsson *et al*, 2011). Early paclitaxel treatment as monotherapy led to a modest reduction in growth of mammary tumors and lung metastases (Fig 6H–J). We found that, similar to *SPP1* deficiency, *TNC* downregulation led to the sensitization of both mammary tumors and lung metastases to paclitaxel treatment (Fig 6H–J). These findings suggest that the

---

**Figure 5. *SPP1* and *TNC* are induced by chemotherapy via JNK signaling.**

A Gene expression profile of MDA231-LM2 cells treated with paclitaxel (PAX) for 48 h. Highlighted are examples of JNK signature genes that are induced by PAX treatment.

B Venn diagram showing overlap between genes that were significantly ($P < 0.05$) induced more than 1.5-fold in response to paclitaxel treatment of MDA231-LM2 cells and genes that were significantly ($P < 0.05$) induced in MDA231-LM2 cells after formation of oncospheres. Significant fold changes were determined by two-sample empirical Bayer tests with Benjamini–Hochberg correction for multiple testing. PAX, paclitaxel treatment.

C GSEA normalized enrichment scores of four independent mammary stem cell signatures that arise in response to paclitaxel treatment of MDA231-LM2 cells. Signatures a-d are the same as in Fig 2E. *$P < 0.05$, ***$P < 0.001$. NS, not significant. *P*-values were determined by random permutation test.

D Immunofluorescence analysis of p-c-Jun expressing cancer cells in lung metastases from mice injected intravenously with MDA231-LM2 cells and treated with PAX at week 3 post-injection. Lungs were harvested 72 h after PAX treatment. Cancer cells express GFP as a marker. DAPI was used for nuclear staining. Scale bar, 50 μm.

E Quantification of cancer cells expressing p-c-Jun (p-c-Jun+) in panel (D). Values are mean from three mice ± SEM. Nine metastatic foci were analyzed per mouse. *P*-value was determined by a two-tailed Mann–Whitney test.

F Western blot analyzing phosphorylated JNK (p-JNK), JNK, phosphorylated c-Jun (p-c-Jun), and c-Jun in MDA231-LM2 cells treated with PAX for 48 h. Tubulin was used as a loading control.

G, H *SPP1* and *TNC* expression in SUM159-LM1 cells treated with increasing concentrations of PAX or doxorubicin (DOXO). Values represent the mean of qPCR triplicates ± SD.

I *SPP1* and *TNC* expression in SUM159-LM1 cancer cells treated with 4 μM 5-fluorouracil or 4 μM methotrexate. Cells were treated for 48 h. Values represent the mean of qPCR triplicates ± SD.

J *SPP1* and *TNC* expression in SUM159-LM1 cells treated with paclitaxel and JNKi. Values represent the mean of qPCR triplicates ± SD. *SPP1* and *TNC* repression by JNKi, $P < 0.05$ for both vehicle and PAX-treated cells. *P*-values were determined by a Mann–Whitney test.

Source data are available online for this figure.

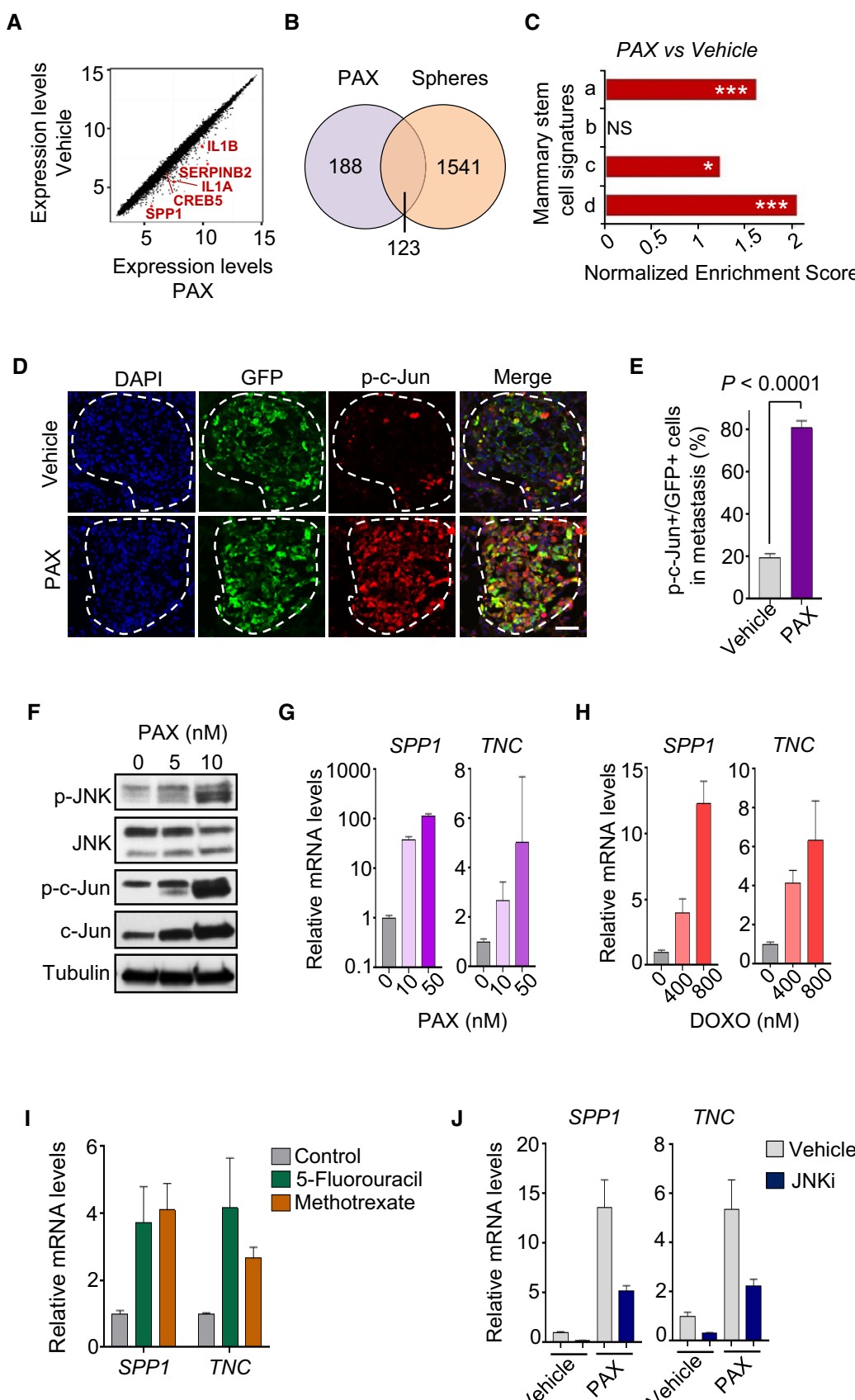

**Figure 5.**

JNK-regulated matricellular proteins SPP1 and TNC promote resistance to chemotherapy.

### Combination treatment with JNK inhibitor and chemotherapy suppresses the progression of metastatic breast cancer

To determine whether inhibition of JNK signaling sensitizes mammary tumors and lung metastases to chemotherapy, we treated tumor-bearing NSG mice with JNKi starting at 5 days post-implantation of MDA231-LM2 or SUM159-LM1 cells and initiated paclitaxel treatment at day 10 post-implantation (Fig 7A). In this setting, single treatment with paclitaxel caused a moderate reduction in mammary tumor growth (Fig 7B). However, a significant improvement in the response was observed by including JNKi to the treatment regimen (Fig 7B). Moreover, we observed a marked reduction of lung metastasis using the combined therapy (Fig 7C and D). The growth-suppressing effect of JNKi and paclitaxel treatment was associated with an increase in cancer cell apoptosis (Fig 7E and Appendix Fig S11). To investigate a potential link between JNK activity and therapy resistance in clinical samples, we examined gene expression data sets from breast cancer patients who had undergone neoadjuvant anthracycline chemotherapy treatment (Li *et al*, 2010). This analysis revealed a significant enrichment of JNK signature genes in patients who responded poorly to the therapy, as compared to those that showed a pathological complete response (pCR), suggesting that JNK activity promotes chemoresistance in patients with breast cancer (Fig 7F). Further GSEA using different cancer therapy resistance gene signatures to assess JNK-mediated transcriptomic profiles also indicated a link between JNK activity and resistance to anti-cancer therapies (Appendix Fig S12). Moreover, breast cancer patients subjected to chemotherapy exhibited poor metastasis-free and overall survival if the JNK signature was enriched within the tumor data set (Li *et al*, 2010; Curtis *et al*, 2012; Fig 7G and H). These results indicate that the JNK pathway, which is active in metastatic breast cancer cells and further induced upon exposure to chemotherapy, limits treatment efficacy. Combining chemotherapy with JNK inhibition may therefore be considered to improve therapeutic responses in metastatic breast cancer.

## Discussion

In this study, we show that activation of the stress-induced JNK signaling pathway promotes breast cancer progression to metastasis by upregulating SPP1 and TNC, two ECM proteins that normally reside within stem cell niches. We find that this same molecular network is induced in cancer cells upon chemotherapy treatment and primes tumors to resist the cytotoxic effects of the treatment (Fig 7I). Thus, these parallel molecular events induced by chemotherapy significantly weaken its efficacy and reduce therapeutic benefits.

Our results indicate that JNK activity is heterogeneous in populations of cancer cells within breast tumors. In metastatic nodules, the proportion of cancer cells exhibiting active JNK signaling is high in micrometastases but decreases as the nodules progress. These findings suggest that a portion of cancer cells in secondary organs lose JNK activity as the metastasis grows, a process that may be explained by a hierarchical stem cell model. This is consistent with previously reported evidence from single cell analysis of metastatic nodules in patient-derived xenografts, that demonstrated an enrichment of a stem-like gene expression signature early during colonization, which was proportionally reduced upon metastatic growth (Lawson *et al*, 2015). Indeed, our findings reveal activation of a mammary stem cell program by JNK signaling, supporting the notion that metastatic cancer cells in early colonization are enriched in stem cell properties. Importantly, our results show that macrometastases, that are heterogeneous with respect to JNK signaling, can re-establish broad JNK activity upon chemotherapy treatment. Thus, chemotherapy may cause increased JNK activity and consequently higher prevalence of cancer cells with stem cell properties in primary tumors and metastases.

---

**Figure 6.** *SPP1* and *TNC* promote chemoresistance.

A  Experimental treatment schedule of mice after cancer cell implantation in the mammary gland to address the role of SPP1 in resistance to PAX.
B  Growth curves of mammary tumors in vehicle- or PAX-treated *Spp1*$^{+/-}$ mice implanted with the indicated cells expressing control shRNA, and *Spp1*$^{-/-}$ mice implanted with the indicated cells expressing *SPP1* shRNA. MDA231-LM2: control, PAX, and SPP1 def. + PAX *n* = 16 tumors per group, SPP1 def. *n* = 14 tumors. SUM159-LM1: control, PAX, and SPP1 def. + PAX *n* = 16 tumors per group, SPP1 def. *n* = 15 tumors. Each value represents the mean ± SEM. ****$P$ < 0.0001.
C  Quantification of lung metastasis from the animals described in panels (A and B). The number of metastatic foci per field of view (FOV) or per lung section (LS) was determined. MDA231-LM2 tumor-bearing mice: control, PAX, and SPP1 def. + PAX *n* = 8 mice per group, SPP1 def. *n* = 7 mice. SUM159-LM1 tumor-bearing mice: control, PAX, and SPP1 def. + PAX *n* = 8 mice per group, SPP1 def. *n* = 7 mice. Ten fields per lung section were quantified. Each value represents the mean ± SEM.
D  Tumor weight in mice at day 40 after implantation of MDA231-LM2 breast cancer cells in a control or SPP1 deficient setting undergoing combination treatment with doxorubicin (adriamycin) and cyclophosphamide (AC regimen). Control, AC, and SPP1 def. *n* = 16 tumors per group, SPP1 def. + AC *n* = 14 tumors. Boxes represent median with upper and lower quartiles. Whiskers show maximum and minimum.
E  Lung metastasis was quantified in mice that were implanted with MDA231-LM2 tumors and subjected to AC chemotherapy. Each value represents the mean ± SEM; *n* = 5 mice per group. Ten random fields per lung section were analyzed for percentage of metastatic area.
F  Representative histological images of metastatic nodules in lungs of mice from the experiment described in panel (E). Metastases were determined by expression of human vimentin (arrows). Scale bar, 100 μm.
G  Treatment schedule after cancer cell implantation to address the role of TNC in PAX resistance in a xenograft mouse model for breast cancer metastasis to lungs. Mice were implanted with MDA231-LM2 cells expressing either control shRNA (control) or a *TNC* shRNA (shTNC), treated with paclitaxel or vehicle.
H  Mammary tumor growth curves in PAX-treated NSG mice bearing control or TNC-knockdown MDA231-LM2 tumors. Values are means ± SEM, *n* = 16 tumors per group. ****$P$ < 0.0001.
I  Lung metastases quantified in tumor-bearing mice from panel (H). Values are means ± SEM, *n* = 16 tumors per group.
J  Representative histology of human vimentin-positive lung metastases (arrows) from the experiment described in panels (G–I). Scale bar, 100 μm.

Data information: *P*-values in panels (B–E, H, and I) were determined by two-tailed Mann–Whitney test.

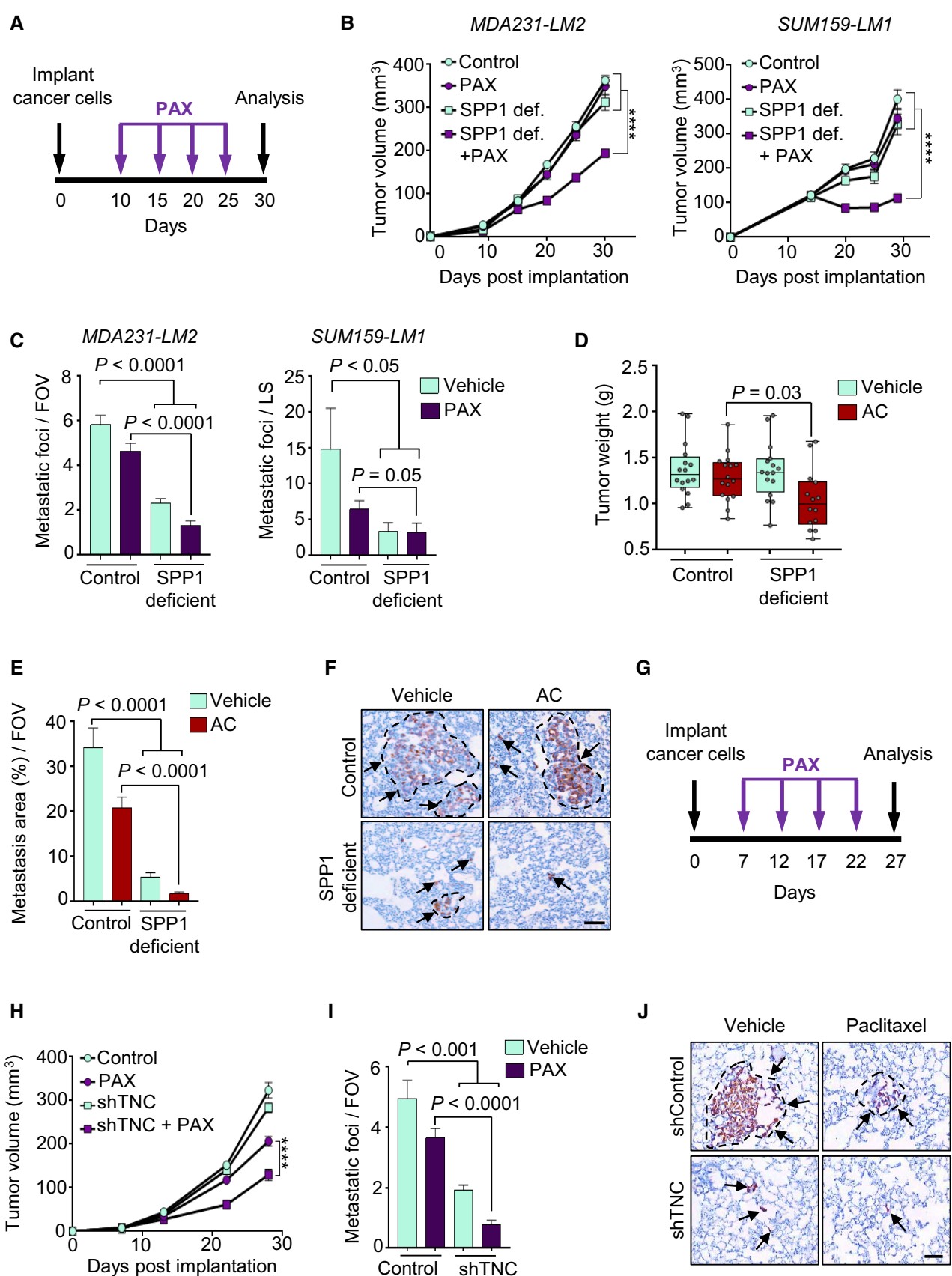

**Figure 6.**

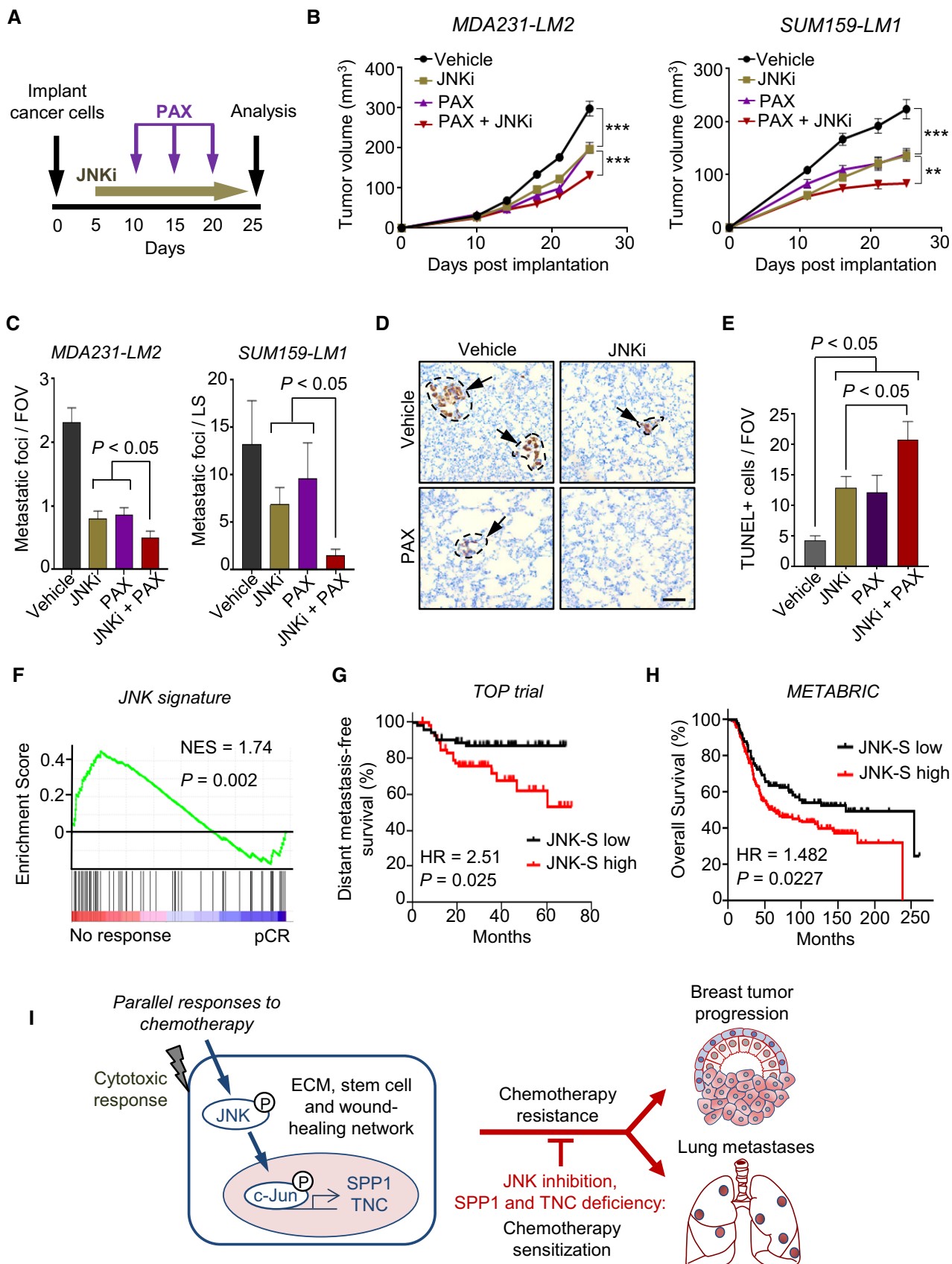

**Figure 7.**

**Figure 7.  Inhibition of JNK signaling sensitizes mammary tumors and metastases to chemotherapy.**

A    Experimental scheme to determine whether JNK signaling promotes resistance to chemotherapy. MDA231-LM2 or SUM159-LM1 cells were implanted bilaterally into the fourth mammary fat pads of NSG mice. Treatment with JNKi was initiated at day 5 post-implantation and repeated every 3 days; paclitaxel chemotherapy was started at day 10 post-implantation and repeated every 5 days.

B    Mammary tumor growth curves in NSG mice implanted with MDA231-LM2 or SUM159-LM1 cells and treated with paclitaxel as single agent or in combination with JNKi. Each value represents the mean ± SEM. MDA231-LM2: *n* = 12 tumors per group. SUM159-LM1: vehicle, JNKi, and PAX *n* = 12 tumors per group; PAX + JNKi *n* = 10 tumors. **\*\*P* < 0.01, \*\*\*P* < 0.001. *P*-values were determined by two-tailed Mann–Whitney test.

C    Metastatic burden in lungs of MDA231-LM2 and SUM159-LM1 tumor-bearing mice treated with paclitaxel as single therapy or in combination with JNKi. Each value represents the mean ± SEM. MDA231-LM2 tumor-bearing mice: *n* = 6 mice per group. SUM159-LM1 tumor-bearing mice: vehicle, JNKi, and PAX *n* = 6 mice per group; PAX + JNKi *n* = 5. Ten fields were quantified per lung section. FOV, field of view; LS, lung section; PAX, paclitaxel. *P*-values were determined by two-tailed Mann–Whitney test.

D    Representative histological images of metastatic foci (arrows) in lungs of the MDA231-LM2-implanted mice described in panel (C), stained for human vimentin expression in differentially treated groups. Scale bar, 100 μm.

E    Apoptosis was determined by quantification of TUNEL-positive cells per field of view (FOV) in MDA231-LM2 tumors (from the animals described in (C)) treated with JNKi and/or PAX. Values are means from six mice per condition ± SEM. *P*-values were determined by two-tailed Mann–Whitney test.

F    GSEA showing enrichment of the JNK signature in tumor samples from breast cancer patients from the TOP trial (GSE16446), stratified according to pathological response to neoadjuvant chemotherapy. NES, normalized enrichment score. pCR, pathological complete response. *P*-value was determined by random permutation test.

G, H    Kaplan–Meier analyses of chemotherapy-treated breast cancer patients associating JNK signature (JNK-S) with metastasis-free survival (TOP trial data set, JNK-S low *n* = 54 patients; JNK-S high *n* = 53 patients (G)) or overall survival (METABRIC data set, JNK-S low *n* = 135 patients; JNK-S high *n* = 135 patients (H)). HR, hazard ratio. *P*-values were determined by log-rank test.

I    Model illustrating the link between stress-induced ECM, stem cell, and wound healing network and metastatic progression and therapy resistance in breast cancer.

The outcome of JNK signaling in cancer cells can vary significantly and is highly context-dependent. Studies on the onset of mouse mammary tumor formation suggest for instance that JNK may play a tumor-suppressive role under certain conditions (Chen *et al*, 2010; Girnius *et al*, 2018a). This may be of a particular relevance at the stages of cellular transformation and early tumorigenesis. Moreover, the molecular differences between breast cancer subtypes and the composition of genetic mutations are likely to affect the outcome of sustained JNK signaling in cancer cells. Exploration of the somatic mutations observed in breast cancer patients with distinct subtypes may provide some insight in this regard. For example, loss-of-function mutations in *MAP3K1* and *MAP2K4*, two kinases involved in activation of JNK, are most commonly found in luminal breast cancer of which approximately 15% contain mutations in either gene (Cancer Genome Atlas Network, 2012). Furthermore, in luminal breast tumors, a notable overlap is observed between inactivating *MAP3K1* or *MAP2K4* mutations and activating mutations in the *PIK3CA* gene that codes the catalytic subunit (p110α) of phosphatidylinositol-3-kinase (PI3K; Ellis *et al*, 2012). Interestingly, evidence suggests that these mutations in *MAP3K1* and *PIK3CA* may provide a cooperative benefit to luminal breast cancer cells via enforced PI3K-AKT signaling (Avivar-Valderas *et al*, 2018). However, in basal-like breast cancer, mutations in *MAP3K1* or *MAP2K4* are notably absent (Cancer Genome Atlas Network, 2012), suggesting that they may not provide similar increase in cellular fitness to what is observed in luminal breast cancer cells. Further research will be needed to elucidate the details of the context dependency of JNK signaling at different stages of cancer progression and across different breast cancer subtypes.

The basal-like subtype of breast cancer is recognized to display stem cell features. Notably, our analysis of tumor samples from a large cohort of breast cancer patients suggested a link between basal-like breast cancers and high JNK activity. This is in concordance with our results showing JNK-induced mammary stem cell properties in breast cancer cells promoting metastatic progression and therapy resistance. Moreover, this suggests that basal-like breast tumors may have an intrinsic ability to evade JNK-mediated apoptosis. Interestingly, *TP53* mutations are particularly frequent in basal-like breast cancer and can be found in over 80% of cases, whereas in non-basal tumors, they are significantly less common (Sorlie *et al*, 2001; Cancer Genome Atlas Network, 2012). Furthermore, basal-like breast tumors appear to be particularly prone to exhibit induction of the nuclear factor kappa-light-chain-enhancer of activated B-cell (NF-κB) pathway compared to other subtypes (Yamaguchi *et al*, 2009; Wang *et al*, 2012). NF-κB has been shown to promote cell survival via induction of anti-apoptotic proteins such as B-cell lymphoma 2 (Bcl-2) family member Bcl-xL (Chen *et al*, 2000; Greten *et al*, 2004). Thus, inactivation of p53 and/or activation of the NF-κB pathway may be required for developing basal-like breast cancers to benefit from JNK pathway activation. Our work provides rationale for combining JNK inhibition and chemotherapy against metastatic breast cancer, and these results may be of particular importance when considering basal-like breast cancer, against which few targeted therapeutic options are available beyond classical chemotherapy regimens.

Cancer cells may escape therapeutic intervention via a number of distinct mechanisms that are often linked to stem cell properties. This includes expression of ATP-binding cassette transporters for drug efflux, enhanced DNA repair capacity and the plasticity to adopt a quiescent state (Batlle & Clevers, 2017). We show how breast cancer cells with stem cell features can induce ECM proteins to promote therapy resistance. The ECM is increasingly recognized as an important player in cancer progression, and its involvement is diverse and extends greatly beyond only providing a structural scaffold (Hynes, 2009). Evidence from gene expression profiles has revealed a number of ECM components that are present within gene signatures of tumor stroma and are associated with poor clinical outcome (Farmer *et al*, 2009; Calon *et al*, 2015). In addition, proteomic analysis of xenograft mammary tumors has shown that tumors with high metastatic potential produce distinct ECM components (Naba *et al*, 2014). At the metastatic site, fibronectin is induced in pre-metastatic niches and matricellular proteins, such as TNC and POSTN, have been shown to be essential constituents of the metastatic niche in lungs (Kaplan *et al*, 2005; Tavazoie *et al*, 2008; Malanchi *et al*, 2011; Oskarsson *et al*, 2011). Moreover, SPP1 has been shown to induce cellular motility in breast cancer cells and

to be a member of a gene set that mediates breast cancer metastasis to the bones (Tuck et al, 2000; Kang et al, 2003). ECM remodeling enzymes like lysyl oxidase can induce collagen cross-linking and promote metastatic progression (Erler et al, 2006). However, the role for ECM components in the response or resistance to cancer therapy has remained largely unclear. In the present study, we provide evidence indicating that ECM proteins can play an important functional role in resistance to therapy. Our results reveal that SPP1 and TNC are directly induced by the c-Jun transcription factor in breast cancer cells in response to stress and serve as essential mediators of chemotherapy resistance. Inhibition of JNK activity provides an opportunity to repress simultaneously the expression of SPP1 and TNC in the cancer cells. Future studies may reveal whether the same mechanism also applies when SPP1 and TNC are produced by the cancer stroma.

Analysis of the JNK-mediated transcriptome in breast cancer cells revealed a significant induction of genes involved in wound healing and tissue regeneration. For example, SPP1 and TNC are generally highly induced during involution of the mammary gland, a process that is associated with major tissue remodeling (Jones et al, 1995; Rittling & Novick, 1997; Schedin, 2006). A recent study showed that JNK signaling plays a significant role in mammary gland involution (Girnius et al, 2018b). In the light of this and our own results, it is intriguing to note that ECM isolated from involuting mammary glands has been shown to support tumor progression and metastasis (McDaniel et al, 2006; Lyons et al, 2011).

The molecular context generated by the tumor microenvironment is an important determinant of cancer cell fitness. The microenvironment has gained increased attention as an important regulator of cancer progression, and growing evidence indicates that signals from the stroma promote cancer cell survival under therapeutic treatment (Gilbert & Hemann, 2010; DeNardo et al, 2011; Shree et al, 2011; Acharyya et al, 2012; Nakasone et al, 2012). Therefore, the combination of inhibiting cancer cell viability/growth and targeting the supportive microenvironment—the niche—may be essential to maximize therapeutic efficacy. Whereas chemotherapy can deliver significant benefits to cancer patients, it remains rarely curative against metastatic cancers. The lack of specificity may cause unintended and counteractive effects. Chemotherapy has been shown to promote cancer cell dissemination in experimental models by modifying the tumor microenvironment (Karagiannis et al, 2017). However, the understanding of general cancer cell responses to chemotherapy is still rudimentary. Our results place JNK signaling at the center of these events where a JNK response in cancer cells is induced by chemotherapy. These signals mediate, among many targets, specific ECM components that play a major role in chemoresistance. Considering ECM proteins as a potential source of drug targets within the cancer microenvironment may provide an opportunity to expand treatment approaches.

# Materials and Methods

## Cell culture

MDA231-LM2 cells, provided by Joan Massagué (Minn et al, 2005), were cultured in DMEM GlutaMAX supplemented with 10% vol/vol fetal bovine serum (FBS), 50 IU/ml penicillin, 50 µg/ml streptomycin,

and 1 µg/ml amphotericin B (Pen/Strep/AmpB). 1-7HB2 cells (Sigma-Aldrich) were maintained in the same media as MDA231-LM2 cells with addition of 5 µg/ml hydrocortisone and 10 µg/ml human insulin. SUM159 cells (Asterand Bioscience) and their lung metastatic derivatives, SUM159-LM1 cells, were cultured in DMEM/F12 medium supplemented with 5% vol/vol FBS, 1 µg/ml hydrocortisone, 5 µg/ml human insulin, and Pen/Strep/AmpB. Primary human mammary epithelial cells (HMEC, Lonza) were cultured in MEGM medium supplemented with the Bullet Kit (Lonza), according to manufacturer's instructions. We maintained primary pleural effusion and ascites samples from metastatic breast cancer patients in a 1:1 mix of supplemented M199 medium (Gomis et al, 2006) and modified M87 medium (DeRose et al, 2013; details in Appendix Supplementary Methods).

## Pleural effusion and ascites samples

Pleural effusion and ascites samples were obtained from breast cancer patients admitted to the Department of Gynecology, University Clinic Mannheim, University of Heidelberg, and the National Center for Tumor Diseases Heidelberg (NCT). The studies were approved by the ethical committee of the University of Mannheim (case number 2011-380N-MA) and the University of Heidelberg (case number S-295/2009) and conformed to the principles set out in the WMA Declaration of Helsinki and the Department of Health and Human Services Belmont Report. Written informed consent was obtained from all patients. Samples were processed as previously described (Gomis et al, 2006). Briefly, pleural effusion or ascites fluids were centrifuged at 300 g for 5 min. Cell pellets were lysed using ACK red blood cell lysis buffer (Lonza) according to manufacturer's instructions. Cells were washed with PBS (Sigma-Aldrich) and plated for culture.

## Oncosphere cultures

Cancer cells were grown as spheroids in ultra-low-attachment flasks (Corning), plated at a density of 25,000 cells/ml in HuMEC medium (Invitrogen), and supplemented with 5 µg/ml human insulin, 20 ng/ml EGF, 10 ng/ml basic fibroblast growth factor (bFGF, Invitrogen), and 2% vol/vol B27 (Life Technologies).

## *In vitro* drug treatments

For gene expression studies involving chemotherapy treatment, breast cancer cells were treated with 0–50 nM paclitaxel (LC Labs), 0–800 nM doxorubicin (LC Labs), 4 µM 5-fluorouracil (Sigma-Aldrich), or 4 µM methotrexate (Santa Cruz) for 48 h. Treatment with 0.1% DMSO was used as vehicle control. JNK inhibitor (CC-401, Santa Cruz) was used to address JNK-dependent gene expression induced by chemotherapy. SUM159-LM1 cancer cells were pre-treated with 4 µM CC-401 or vehicle (0.1% DMSO) for 24 h. Then, 10 nM paclitaxel or vehicle (0.1% DMSO) was added to each of the pre-treated cultures. Cells were treated with chemotherapy for 48 h and analyzed. CC-401 was furthermore used to determine JNK-dependent gene expression induced in oncospheres. SUM159-LM1 cells (monolayer) were pre-treated with 4 µM CC-401 or vehicle (0.1% DMSO) for 48 h. Cells were trypsinized and seeded as spheroids in the presence of 4 µM CC-401 or vehicle (0.1% DMSO). Gene expression in the oncospheres was analyzed after 7 days of culture.

### Expression of active JNK

We obtained MKK7B2Jnk1a1 and MKK7B2Jnk1a1(APF) cDNA from the expression vectors pCDNA3 Flag MKK7B2Jnk1a1 (Addgene plasmid #19726), and pCDNA3 Flag MKK7B2Jnk1a1(APF) (Addgene plasmid #19730), both provided by Roger Davis, and subcloned into the pLVX-Puro lentiviral vector via BstBI and XmaI restriction sites.

### Generation of SPP1 and TNC-knockdown cells

*SPP1* knockdown was generated with either GIPZ (GE Dharmacon) or miR-E lentiviral vectors (Fellmann *et al*, 2013) expressing shRNA against SPP1 gene products. miR-E SPP1 hairpins were produced from the StagBFPEP lentiviral vector, a modified version of the original SGEP vector kindly provided by Johannes Zuber (IMP—Research Institute of Molecular Pathology GmbH, Vienna) in which the constitutively expressed GFP protein is replaced by the tagBFP protein. TNC knockdown was generated with Mission TRC (Sigma-Aldrich) against TNC mRNAs.

### Matrigel invasion assays

We performed invasion assays using 24-well Matrigel invasion chambers (BD Biosciences) according to manufacturer's instructions. MDA231-LM2 or SUM159-LM1 cells were pre-treated for 48 h with 4 μM CC-401 or 0.1% DMSO (Sigma-Aldrich) as a vehicle control. Then, 100,000 cells were seeded on a transwell in the presence of JNK inhibitor or DMSO and were allowed to invade through the Matrigel coating for 20 h. Invading cells were then stained with 0.2% vol/vol crystal violet (Sigma-Aldrich) + 2 mM glycine (Sigma-Aldrich) and washed with distilled water. Inserts were mounted on slides, and cells were counted under a light microscope (Zeiss Primovert inverted microscope).

### Western blot

PVDF membranes (Bio-Rad) blotted with proteins from whole cell lysates were probed with antibodies against c-Jun (1:1,000), JNK (1:1,000), p-c-Jun (Ser63) (1:500), p-JNK (Thr183/Tyr185; 1:500, all from Cell Signaling), tenascin C (1:1,000, Santa Cruz), GAPDH (1:10,000, Sigma-Aldrich), or tubulin (1:10,000, Sigma-Aldrich). Following primary antibody incubation, membranes were washed and probed with anti-mouse IgG-HRP (1:10,000, Leica) or anti-rabbit IgG-HRP (1:10,000, Leica) and exposed to X-ray films (Fujifilm) or imaged in a ChemiDoc imaging system (Bio-Rad).

### ChIP

ChIP assays were performed using the Pierce™ Magnetic ChIP Kit (Thermo Scientific) according to manufacturer's instructions. Briefly, 4–5 × 10$^6$ MDA231-LM2 or SUM159-LM1 cells were treated with 1% formaldehyde for 10 min at room temperature, followed by glycine during 5 min at room temperature for chromatin–protein cross-linking. After PBS wash, cells were collected in PBS supplemented with HALT protease/phosphatase inhibitor cocktail, using a cell scraper. Cell nuclei were extracted using membrane extraction buffer containing protease/phosphatase inhibitors (10 min incubation on ice). DNA–protein complexes were digested with

microccal nuclease (MNase, 8 U/μl) for 15 min at 37°C. Following MNase mediated digestion, nuclear membranes were ruptured by 3 × 20 s pulses of sonication, on ice, using a W-250 D digital microtip sonifier (Branson). 10% of each sample was used as input reference control. Chromatin–protein suspensions were then incubated with 10 μg of control rabbit IgG isotype antibody or a c-Jun antibody (both from Cell Signaling) overnight at 4°C. Antibody-bound chromatin–protein complexes were isolated by Protein A/G Magnetic Beads and digested with Proteinase K for 1.5 h at 65°C. DNA Clean-Up columns (Thermo Scientific) were used to purify DNA in samples, according to manufacturer's instructions. Eluted DNA was then submitted to SYBR green (Applied Biosystems) qPCR analysis using the Viia 7 Real-Time PCR System (Applied Biosystems).

### Immunohistochemistry

Tissues derived from mouse studies were fixed in formalin overnight at 4°C and mounted in paraffin using the Tissue-Tek VIP 6 Vacuum Infiltration Processor (Sakura). Sections (5–8 μm) were cut using a Microm HS355S microtome (Thermo Fisher Scientific) and incubated at 56°C for 1 h. For fixed-frozen sections, formalin-fixed tissues were embedded in OCT compound (Sakura) and frozen at −80°C. Sections (8–10 μm) were cut using a Microm HM 525 cryotome (Thermo Fisher Scientific). For vimentin staining of mouse lung tissues or phosphorylated c-Jun (Ser73) staining of patient-matched primary tumor and lung metastasis samples, paraffin- or OCT-embedded sections were rehydrated, treated with 3% hydrogen peroxide for 10 min and antigen retrieval was performed using citrate buffer (pH 6.0, Vector Laboratories) at 100°C for 20 min. Then, sections were incubated with blocking buffer, followed by incubation with respective primary antibodies (human vimentin: Leica, clone SRL33, 1:400; p-c-Jun (Ser73): Cell Signaling, clone D47G9, 1:50). Corresponding anti-mouse or anti-rabbit IgG biotinylated secondary antibodies and an ABC avidin–biotin–DAB detection kit (all from Vector Laboratories) were used for detection and visualization of antigens according to manufacturer's instructions. Sections were analyzed using a Zeiss AxioPlan microscope.

### Tissue microarray analysis

A breast carcinoma tissue microarray (TMA) consisting of clinically annotated cases with survival data was purchased from US Biomax (HBre-Duc140Sur-01). The TMA slide containing sections of biopsy cores of breast tumors from patients were processed for immunohistochemistry as described in the previous section, but stained using primary antibodies against phosphorylated JNK (Thr183/Tyr185, clone 81E11, Cell Signaling, 1:50). Following staining, the TMA was scanned and analyzed using the Image Scope (version 11.2.0.780) software. All samples from patients that progressed and died from the disease were analyzed. Evaluation of p-JNK high or low expression in biopsy cores was performed under the supervision of a pathologist.

### Immunofluorescence

Murine lung tissues were fixed in formalin overnight at 4°C, washed with PBS, and incubated in 30% sucrose overnight. Tissues were embedded in OCT compound (Sakura) and frozen at −80°C. Sections (8–10 μm) were cut using a Microm HM 525 cryotome

(Thermo Fisher Scientific), air-dried for 2 h at room temperature, and washed with PBS. Non-specific binding was blocked using TNB buffer (0.1 M Tris–HCl + 0.15 M NaCl + 0.5% blocking reagent, PerkinElmer) for 1 h at room temperature. Antibodies against phosphorylated c-Jun (Ser63, clone KM-1, Santa Cruz) and GFP (Abcam) were diluted in TNB buffer (1:100 and 1:1,000, respectively) and incubated on tissue sections at 4°C overnight. Sections were washed with PBS + 0.05% Tween 20 (PBS-T), followed by incubation of secondary anti-mouse-Cy3 or anti-chicken-Alexa488 antibodies (Invitrogen) in TNB buffer for 1 h at room temperature. Slides were mounted and analyzed using a Zeiss Cell Observer microscope.

## Mouse studies

Animal care and all procedures were previously approved by the governmental review board of the state of Baden–Wuerttemberg, Regierungspraesidium Karlsruhe, under the authorization numbers G50/13, G81/15, G81/16, and G289/16 and followed the German legal regulations. Non-obese diabetic-severe combined immunodeficiency gamma (NSG) female mice of 6–8 weeks of age were generally used for mouse studies, except for experiments involving osteopontin deficiency in both cancer cells and stroma. For these studies, Spp1 knockout mice (B6.129S6(Cg)-Spp1tm1Blh/J, JAX stock #004936 The Jackson Laboratory (Liaw et al, 1998)) were crossed with NSG mice (The Jackson Laboratory) to generate either $Spp1^{+/-}$ Scid $Il2rg^{null}$ (control) or $Spp1^{-/-}$ Scid $Il2rg^{null}$ (SPP1 deficient), of which female littermates were used for the experimental studies. Mice were housed in individually ventilated cages under temperature and humidity control. Cages contained an enriched environment with bedding material.

For spontaneous lung metastasis assays, we injected either MDA231-LM2 or SUM159-LM1 cells bilaterally into the fourth mammary fat pads of anesthetized mice (isoflurane). 500,000 MDA231-LM2 cells or 250,000 SUM159-LM1 cells were suspended in a 1:1 (vol/vol) mixture of Matrigel and PBS and inoculated into mammary fat pads, in a volume of 50 μl. Primary tumor growth was monitored by regular measurements using a digital caliper. After indicated times (25–40 days post-cancer cell injection), mice were sacrificed and metastatic burden in the lungs was determined by either ex vivo imaging or analysis of histological sections using human-specific vimentin antibody (Leica, clone SRL33, 1:400). Lung colonization assays were performed by injection of 100,000 MDA231-LM2 or SUM159-LM1 cells in the lateral tail vein of NSG mice. Injected cancer cells were previously transduced with a triple reporter expressing the genes herpes simplex virus thymidine kinase 1, green fluorescent proteins (GFP), and firefly luciferase (Ponomarev et al, 2004). For ex vivo imaging, mice were injected intraperitoneally with 150 mg/kg body weight D-luciferin (Biosynth) prior to sacrifice and then lungs were imaged with IVIS Spectrum Xenogen machine (Caliper Life Sciences). Bioluminescent analysis was performed using Living Image software, version 4.4 (Caliper Life Sciences). To study tumor initiation capacity of breast cancer cells, we implanted 2,500, 1,000, or 500 MDA231-LM2 cells subcutaneously into NSG mice and monitored tumor formation. The cancer cells were implanted in a 50 μl mix of PBS and Matrigel (1:1). At day 32 post-cancer cell implantation, mice were sacrificed and tumor formation was recorded.

## In vivo drug treatments

For mammary fat pad and lung colonization assays, tumor-bearing NSG mice were injected intraperitoneally with CC-401 (25 mg/kg; AdooQ Bioscience) or vehicle control (3% DMSO in PBS) every 3 days. For tumor initiation experiments, MDA231-LM2 cells were pre-treated in vitro for 48 h with 4 μM JNK inhibitor (CC-401) or a vehicle control before subcutaneous implantation in mice. The mice were treated with CC-401 (25 mg/kg) or a vehicle saline-based 3% DMSO control. Treatment was repeated every third day. For chemotherapy experiments, mice were treated with a combination of doxorubicin (1 mg/kg) and cyclophosphamide (30 mg/kg; Sigma-Aldrich) or paclitaxel (15 mg/kg) at indicated time points. 1% DMSO in PBS was used as vehicle control for doxorubicin and cyclophosphamide experiments and 4% ethanol, cremophor mix (1:1) in saline (0.9% NaCl) was used as control for paclitaxel experiments.

## Gene expression analysis in microarray data sets

RNA samples were hybridized onto Affymetrix GeneChip® Human Genome U133 Plus 2.0 microarrays. Raw gene expression data were RMA-normalized, and two-sample empirical Bayes tests with Benjamini–Hochberg correction for multiple testing were performed using Chipster (Kallio et al, 2011). The following gene expression data sets were used: GSE98265 (includes all data sets generated by the authors), METABRIC (Curtis et al, 2012), GSE16446, GSE14020, GSE2603, and GSE5327.

For the generation of a JNK signature, we retrieved genes that were at least 1.4-fold upregulated in cells expressing MKK7-JNK as compared to empty vector control and that were downregulated at least −1.4-fold in cells exposed to JNK inhibitor as compared to untreated cells (vehicle). Gene set enrichment analysis (GSEA) was performed as previously described (Subramanian et al, 2005). Briefly, all genes were ranked on the basis of their correlation with the respective condition. Thereafter, genes from published gene sets or from our established JNK signature were aligned to the ranked list. An enrichment score was calculated, indicating whether the members of the signature are randomly distributed or primarily found at the top or bottom of the ranked list. Nominal P-values were calculated on the basis of random permutations of the phenotype labels or the gene set, dependent on the number of samples per phenotype. Nominal P-values ≤ 0.05 were considered statistically significant. Gene Ontology analysis of the top 500 upregulated genes in active JNK-expressing cells compared to empty control and the top 500 downregulated genes after treatment with JNK inhibitor compared to vehicle was carried out using Database for Annotation, Visualization and Integrated Discovery (DAVID; Huang et al, 2009a,b). Enriched GO terms with Benjamini–Hochberg-corrected P-values < 0.05 and FDR < 0.1 were regarded as statistically significant.

To classify patient data of these data sets according to our established JNK signature, expression values of the 68 genes were retrieved and normalized within each gene before the mean was calculated. METABRIC and TOP trial data sets and GSE14020-GPL570 metastases cohort contained all 68 genes that were used to classify each patient sample. GSE14020-GPL96 cohort was analyzed according to 58 signature genes since 10 JNK-induced genes were not present on this platform. Median cutoff was used to separate JNK low and high. To address SPP1 and TNC in patient data sets,

mean normalized SPP1 and TNC expression were calculated within each sample and median used to classify patients in SPP1 and TNC low or high groups.

### Statistical analysis

For Kaplan–Meier analyses of breast cancer patients, statistical differences in survival curves were calculated by log-rank (Mantel–Cox) test. To analyze the putative association between JNK activity in tumor samples on TMA and different clinical parameters, binomial test was used. This was done to address whether patient age, tumor stage, or receptor status were potential confounding variables when association between JNK activity and poor patient survival was analyzed. Both Kaplan–Meier and association analyses were performed using GraphPad Prism version 7.02 for Windows. The GSE14020 gene expression data set was used to study the correlation between *JUN*, *SPP1*, and *TNC* expression in a cohort of 65 metastasis samples from breast cancer patients. Probe sets analyzed were the following: *JUN*: 201464_x_at, *SPP1*: 209875_s_at, *TNC*: 216005_at. Gene expression values for each gene within each individual were associated in a correlation matrix. Then, Pearson correlation coefficient (*r*) and *P*-value were calculated for each comparison using GraphPad Prism version 7.02 for Windows. All other experiments were analyzed by two-tailed unpaired Mann–Whitney tests unless specified otherwise. *P*-values ≤ 0.05 were considered significant.

# Data and software availability

Microarray data generated for this study have been submitted to Gene Expression Omnibus under the accession number GSE98265 (super series that contains experiments GSE98237, GSE98238, GSE98239). https://www.ncbi.nlm.nih.gov/geo/query/acc.cgi?acc = GSE98265.

**Expanded View** for this article is available online.

## Acknowledgements
We would like to thank M. Sprick, M. Milsom, A. Trumpp, M. Bettess, S. Acharyya, and S. Vanharanta for critical reading of the manuscript and helpful discussion. We thank M. Socher and the DKFZ Central Animal Laboratory for advice on animal protocols and husbandry, M. Brom (Light Microscopy unit of the DKFZ Imaging and Cytometry Core Facility) for her technical support, and the microarray unit of the DKFZ Genomics and Proteomics Core Facility for providing transcriptomic services. The authors are also indebted to K. Decker and L. Wiedmann for technical assistance. M.P. and C.M.L. were supported by a scholarship from the Helmholtz International Graduate School for Cancer Research. T.H. has been funded by the International Tenure Track Program of the University of Tsukuba, and E.D.B. was supported by a DKFZ Postdoctoral Fellowship. This work was supported by the Dietmar Hopp Foundation, Marie Curie Career Integration Grant (334563 to T.O.), and DFG research grant (AOBJ:633977 to T.O.).

## Author contributions
JI-R and TO designed experiments, analyzed data, and wrote the manuscript. JI-R carried out experiments. MP conducted gene expression analyses and helped with animal experiments. TH performed immunofluorescence stainings and analysis. JM helped with subcloning, cultures of patient samples, and animal experiments. AD and CML helped with generation of knockdown cancer cells and animal experiments. EDB helped with p-JNK analysis. H-PS oversaw pathological evaluation of human breast cancer tissue sections. SS, MS, and AS organized collection of patient samples and provided clinical advice. TO supervised the research. All authors read and discussed the manuscript.

## Conflict of interest
The authors declare that they have no conflict of interest.

### The paper explained

**Problem**
Metastatic progression and resistance to current treatment options constitute major challenges in the battle against cancer. Whereas chemotherapy has been used against malignancies for several decades, the treatment is rarely curative against metastatic cancers. This underscores the need for a better understanding of cancer cell responses to the treatment and how the cells may acquire therapy resistance.

**Results**
We combine evidence from annotated clinical material and experimental models to reveal how JNK stress signaling links induction of stem cell properties in breast cancer cells with acquisition of chemotherapy resistance. We report that JNK signaling induces expression of the matrix components SPP1 and TNC, directly via c-Jun transcription factor, to promote metastasis to the lungs. This molecular axis also plays an important role in therapy resistance, as it is induced by chemotherapy and thus impairs the treatment. Notably, inhibition of JNK signaling or repression of SPP1 or TNC expression can sensitize mammary tumors and lung metastases to chemotherapy.

**Impact**
This study provides novel insights into how cancer cells may benefit from chemotherapy-induced stress signals that counteract cytotoxic responses and limit therapeutic efficacy. Identification of the niche matrix components SPP1 and TNC as relevant mediators of therapy resistance highlights the significance of the ECM as a valuable source of cancer targets. Moreover, our results provide rationale for considering JNK inhibition as a strategy to simultaneously repress the expression of SPP1 and TNC and thus improve therapeutic responses of breast cancer to chemotherapy.

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
