## [Review Process File · EMBO Molecular Medicine]

Stress signaling in breast cancer cells induces matrix components that promote chemoresistant metastasis

Jacob Insua-Rodríguez, Maren Pein, Tsunaki Hongu, Jasmin Meier, Arnaud Descot, Camille M. Lowy, Etienne De Braekeleer, Hans-Peter Sinn, Saskia Spaich, Marc Sütterlin, Andreas Schneeweiss and Thordur Oskarsson.

Review timeline:

Submission date:	18 th February 2018
Editorial Decision:	13 th April 2018
Revision received:	18 th June 2018
Editorial Decision:	12 th July 2018
Revision received:	27 th July 2018
Accepted:	15 th August 2018

Editor: Lise Roth

Transaction Report:

1st Editorial Decision

13th April 2018

Thank you for the submission of your manuscript to EMBO Molecular Medicine. We have now heard back from the three referees whom we asked to evaluate your manuscript.

As you will see from the reports below, while referees 2 and 3 are overall positive and support publication of the article in EMBO Molecular Medicine (pending appropriate revisions), referee 1 mentions the inadequacy of the cell line used (isolated from pleural effusions) and the limitations of the translational applications of your study.

Therefore, an adequate and thorough discussion on these two points, as well as addressing the reviewers' concerns in full will be necessary for further considering the manuscript in our journal, with the exception of additional validation of the JNK signature (referee 1, point 6). EMBO Molecular Medicine encourages a single round of revision only and therefore, acceptance or rejection of the manuscript will depend on the completeness of your responses included in the next, final version of the manuscript.

REFeree REPORTS.

Referee #1 (Comments on Novelty/Model System for Author):

See text under 6 regarding cell lines used and insufficient description of patient series.

Referee #1 (Remarks for Author):

In this manuscript, the authors report that activation of the JNK pathway results in:

- 1) induction of ECM, wound healing response and stem cell properties.
- 2) a worse prognosis of breast cancer patients.
- 3) critical involvement in tumor initiation and metastasis in xenograft mouse models.

4) conferring resistance to chemotherapy.

General remarks:

We know from literature that the JNK pathway has been implicated in many different processes that might be related to tumor initiation, progression and metastasis, but also in central process in each and every cell. Obviously, any stressed cell is prone to activate the JNK pathway, being upstream (cause), downstream (consequence), not directly related to the perturbations done (epiphenomenon) or even mixed. In addition, it is not easy to pinpoint which of the observed phenomena in the different systems used (human samples, tumoroids, cell lines, xenografts, ...) can be mechanically attributed to JNK pathway activation.

Therefore, the experimental findings need a very cautious interpretation throughout the manuscript. Yet, quite a few observations are interpreted as being in line with the hypothesis, but robust validation is partly lacking. In addition, some essential info is missing to put the results into perspective.

Comments and questions:

1. Fig. 1A shows a KM-curve. Essential information about subtypes, treatment, etc. are missing, but should be provided.
2. The differences in IHC shown in Fig. 1B are not sufficiently convincing. It should be taken into account that the size and 'age' of the tumor is different, that just slight differences in the fixation might be enough to cause such differences. For example, many researchers fix the mammary fat pads as a whole before sectioning. Formalin does not penetrate very well fatty tissue.
3. The cell line used is derived from a triple negative breast cancer isolated from pleural effusions. It is highly uncommon for triple negative or basal-like breast cancer to develop pleural effusions like e.g. invasive lobular cancer. TN/basal-like breast cancer is much more prone to metastasize to e.g. the brain. In addition, interpretation of experimental results on cell lines isolated from pleural effusions is a slippery slope. Moreover, wnt signaling is not a prominent signal in the great majority of breast cancer. In this cell line, it is.
4. It is no surprise that the cell line used is very much dependent on the JNK pathway activation.
5. It is not at all a surprise that the tumoroids are dependent on JNK pathway activation.
6. The activated JNK pathway signature has not been sufficiently validated.
7. One would expect much more activation of the JNK pathway in basal like (or triple negative) high grade breast cancers. The cells from these tumors proliferate much faster than e.g. most ER-positive tumors. This in itself is causing quite some stress, in particular when oxygenation and 'nutrition' is insufficient, necrosis occurs and this necrosis also initiates an inflammatory response.
8. The authors performed a GSEA according to the method as published in 2005. Subsequently, they compared resulting gene lists with some that were already published as well as their own JNK pathway signature. Computational biology is not my primary expertise, but it seems that this approach runs a serious risk of overfitting and a biased focus on processes that have been described before to be involved in tumor initiation, progression, and metastasis, i.e. the usual often non-specific suspects.
9. How is 'stemness' defined?
10. Last but not least, stating that targeting activation of the JNK pathway might be a promising new treatment strategy is way too premature that might generate false hope. It is highly likely that targeting such a central process in the cell will eventually lead to an effective treatment without unacceptable toxic side effects, is not realistic.

Referee #2 (Comments on Novelty/Model System for Author):

This is a high quality and highly translational study.

Referee #2 (Remarks for Author):

In this manuscript Rodriguez et al describe a series of experiments in which they generate evidence to indicate that metastasis and chemotherapy resistance are JNK-dependent events in breast cancer (basal subtype in particular). They go on to show this is mediated via c-JUN induced transcriptional control of the expression of SPP1 and TNC. This is a solid study and the in vitro and in vivo cell line data are corroborated by frequent reference to patient samples and publicly available datasets. The authors make a strong case for considering the inclusion of JNK inhibitors in standard of care

treatment of patients with JNK+++ breast cancers.

There are a few concerns that I would like the authors to consider.

1. It would be helpful to quantitate the data in suppl fig 1A.
2. There are no stats on Fig 4C & D or Suppl Fig 8D
3. The data in Fig 4D & G (intercepts the Y axis very high) would suggest that SPP1 is not as critical to the metastasis phenotype as is TNC1. The authors may wish to comment on this.
4. It was unclear to me whether the data in Fig 5D & E were derived from metastatic foci that had seeded from a primary xenotransplant tumour or whether these were "metastases" that were derived from an iv injection. This should be made clear in the text.
5. Fig 6 C & D. The reduced metastatic activity is likely explained by the reduced tumour cell numbers following the combination of chemo + SPPP1 depletion. This should be stated in the text. Significantly though, it is a very impressive reduction in metastatic activity in the SPP1 deficient cells alone.

Referee #3 (Remarks for Author):

This carefully executed study confirms and strengthens previous papers on the links between JNK stress signalling, the extracellular matrix (ECM) and aspects of tumorigenesis. Stress initiated by chemotherapies exploits these pathways, hence the inhibition of JNK signalling or disruption of the expression of the emerging ECM proteins SPP1 and TNC sensitizes experimental breast cancer/metastasis models to chemotherapy.

The experiments carried out are technically strong and the link between basal-like breast cancer with significant stem-cell like features and high JNK activity is clearly established. The role of SPP1 and TNC as 'mediators' of JNK signalling is not entirely proven. Although there is a clear link to the c-Jun transcription factor and to tumor progression and chemoresistance no mechanisms are established. These proteins have previously been implicated in such events and it would be interesting to know if other, perhaps more novel, ECM proteins were evident in the data analyses and may play similar roles. The in vitro studies here were confined to tumor cell lines but the role of the stroma in tumorigenesis is acknowledged, but not studied. Some discussion of the relative role of these and other ECM proteins of the tumor and stroma in tumorigenesis and metastasis would be helpful, as well as further acknowledgement of previous work and how this study complements what is already known.

p.10 line 13 up - the patients were treated with the JNKi

1st Revision - authors' response

18th June 2018

Point by point responses to reviewers

We would like to thank the reviewers for very insightful comments. We have revised our manuscript accordingly. We have also extended the main methods and included full blot pictures as source data for all Western blots used in the manuscript. Changes in the text are highlighted in the red-lined version of the manuscript. Specific reviewer's points are discussed below.

Referee #1 (Comments on Novelty/Model System for Author):

See text under 6 regarding cell lines used and insufficient description of patient series.

Referee #1 (Remarks for Author):

In this manuscript, the authors report that activation of the JNK pathway results in:

- 1) induction of ECM, wound healing response and stem cell properties.
- 2) a worse prognosis of breast cancer patients.
- 3) critical involvement in tumor initiation and metastasis in xenograft mouse models.
- 4) conferring resistance to chemotherapy.

General remarks:

We know from literature that the JNK pathway has been implicated in many different processes that might be related to tumor initiation, progression and metastasis, but also in central process in each and every cell. Obviously, any stressed cell is prone to activate the JNK pathway, being upstream (cause), downstream (consequence), not directly related to the perturbations done (epiphenomenon) or even mixed. In addition, it is not easy to pinpoint which of the observed phenomena in the different systems used (human samples, tumoroids, cell lines, xenografts, ...) can be mechanically attributed to JNK pathway activation.

Therefore, the experimental findings need a very cautious interpretation throughout the manuscript. Yet, quite a few observations are interpreted as being in line with the hypothesis, but robust validation is partly lacking. In addition, some essential info is missing to put the results into perspective.

Comments and questions:

1. Fig. 1A shows a KM-curve. Essential information about subtypes, treatment, etc. are missing, but should be provided.

We have included a supplementary table with the available relevant clinical information of patient samples used in Fig.1A. The table is included in the Appendix Table S1.

2. The differences in IHC shown in Fig. 1B are not sufficiently convincing. It should be taken into account that the size and 'age' of the tumor is different, that just slight differences in the fixation might be enough to cause such differences. For example, many researchers fix the mammary fat pads as a whole before sectioning. Formalin does not penetrate very well fatty tissue.

We used patient-matched primary tumors and metastases for the comparison of p-c-Jun (from 12 patients). These patients are likely to have primary tumors at different sizes and metastatic nodules also varying in size. The quantification (shown with log₁₀ scale) and paired t test (Fig. 1C) revealed a statistically significant increase in p-c-Jun+ cells / tumor area in the metastases compared to the primary tumors, thus penetrating the difference that may be caused by variable tumor sizes. The presence of adipose tissue in our samples was minor and did not make any impact on the immunostaining. Phosphorylated c-Jun staining was consistent and adipose tissue content did not show a negative correlation with staining intensity or number of p-c-Jun positive cells.

3. The cell line used is derived from a triple negative breast cancer isolated from pleural effusions. It is highly uncommon for triple negative or basal-like breast cancer to develop pleural effusions like e.g. invasive lobular cancer. TN/basal-like breast cancer is much more prone to metastasize to e.g. the brain. In addition, interpretation of experimental results on cell lines isolated from pleural effusions is a slippery slope. Moreover, wnt signaling is not a prominent signal in the great majority of breast cancer. In this cell line, it is.

We used 2 triple negative breast cancer cell lines, MDA231-LM2 and SUM159-LM1, for most functional experiments. The reason for this choice is twofold. Firstly, when analyzing clinical samples we observed that JNK activity is enriched in lung metastases as compared to patient-matched primary tumors (Fig. 1B). Therefore, we used MDA231-LM2 and SUM159-LM1 cells that readily metastasize to the lungs when injected into mice. Secondly, triple negative or basal-like breast cancers are especially prone to metastasize to visceral organs, particularly to the lungs, while this subtype is less prevalent in patients that develop bone metastasis (Rodriguez-Pinilla SM et al Clin Cancer Res 2006, Liedtke C et al J Clin Oncol 2008, Gong Y et al Sci Rep 2017). Accumulation of pleural fluids is not uncommon in patients with visceral metastasis and thus triple negative patients do indeed develop this condition (De Rose YS et al Nat Med 2011, Gomez-Miragaya J et al Stem Cell Reports 2017).

Our functional data are not only based on the use of cancer cells derived from pleural effusions or ascites, but also based on experiments performed using cells originally derived from a human breast tumor. Although the parental line of MDA231-LM2 breast cancer cells (MDA-MB-231) is derived originally from a pleural effusion, the parental line of SUM159-LM1 cells (SUM-159) is derived from a primary tumor (Flanagan L et al Breast Cancer Research and Treatment 2000). Furthermore, we have validated our findings in clinical samples and publicly-available datasets from primary tumors and metastases from patients.

4. It is no surprise that the cell line used is very much dependent on the JNK pathway activation.

We would argue that a dependency on JNK signaling is highly context dependent. For example, neither cell lines that we used (MDA231-LM2 and SUM159-LM1) are dependent on JNK pathway activation for their maintenance in regular tissue culture conditions – that is, cultured as a monolayer, in the presence of serum (in vitro treatment with JNK inhibitor under these conditions did not result in any changes in proliferation or survival – see Fig. EV1 B and C). However, dependency on JNK activity becomes evident under conditions of increased stress (i.e. in vivo, in suspension cultures (oncospheres) or chemotherapy exposure).

5. It is not at all a surprise that the tumoroids are dependent on JNK pathway activation.

Based on the enrichment of JNK signature and induction of p-JNK and p-c-Jun that we observed in sphere cultures (Fig. 3B and C), it seemed indeed logical that cancer cells would potentially benefit from JNK signaling to grow as spheroids.

6. The activated JNK pathway signature has not been sufficiently validated.

We were careful when we developed the JNK signature in this study. First, we only selected the genes that were both significantly induced by ectopically active JNK and significantly repressed in response to treatment with JNK inhibitor. Moreover, target genes within the JNK signature (i.e. SPP1 and TNC) were also modulated by JNK activity in distinct breast cancer cell lines. We validated this in: a) second cell line (SUM159-LM1) and; b) in breast cancer cells isolated from 4 different patients. We used the 68 gene JNK signature to make prediction in different and independent data sets (METABRIC, TOP trial (GSE16446) and a data set from 65 dissected metastases from breast cancer patients (GSE14020)). In all these data sets, the results lined up to our predictions.

7. One would expect much more activation of the JNK pathway in basal like (or triple negative) high grade breast cancers. The cells from these tumors proliferate much faster than e.g. most ER-positive tumors. This in itself is causing quite some stress, in particular when oxygenation and 'nutrition' is insufficient, necrosis occurs and this necrosis also initiates an inflammatory response.

This is an interesting point. Indeed, high proliferation rates, low access to oxygen and nutrients, and cytokines represent sources of cellular stress that may lead to activation of JNK pathway activity in cancer cells. Although not entirely, these factors may possibly explain, at least partially, the increased proportion of JNK-S high tumors in basal-like breast cancer patients as compared to luminal and Her2-enriched (considering that basal-like are more frequently high-grade tumors). However, it is important to note that JNK-S high tumors were not exclusively basal-like, but were also found (although less frequently) in luminal and HER2 positive subtypes.

8. The authors performed a GSEA according to the method as published in 2005. Subsequently, they compared resulting gene lists with some that were already published as well as their own JNK pathway signature. Computational biology is not my primary expertise, but it seems that this approach runs a serious risk of overfitting and a biased focus on processes that have been described before to be involved in tumor initiation, progression, and metastasis, i.e. the usual often non-specific suspects.

GSEA of stem cell signatures was based on the results of GO term analysis showing involvement in wound healing and organ development. The analysis is dependent on the data sets that are currently available and can of course not be said to be exhaustive. However, the positive claims that we make were thoroughly researched and we used several fully independent data sets to make each point. This minimizes the probability of confounding effects.

The link of genes to tumor initiation, metastasis, stem cell properties etc. underscores their central role rather than pointing to a “nonspecific” enrichment. This is further demonstrated by our experimental findings in addition to our in silico analysis. For example, we functionally demonstrate that JNK promotes cell motility, invasion and stem cell properties (Figure EV2, Figure 3F-M) as

predicted by our GSEA and GO analysis. Establishment of the JNK signature and our GSEA on five MaSC gene sets were conducted independently of each other (please see point 6 for details on JNK signature).

9. How is 'stemness' defined?

We define stemness as the acquisition of molecular properties of stem cells, such as increased expression levels of stem cell markers and normal mammary stem cell gene signatures in a population of cells. We used the term stemness on page 8. Since the term may appear rather vague, we have replaced it with the term of “stem cell properties”, which we would also define as the acquisition of proteins and/or other molecular markers that are representative of stem cells. We do not make any claims of cell of origin, but focus on the molecular properties that are induced by JNK signaling in breast cancer cells.

10. Last but not least, stating that targeting activation of the JNK pathway might be a promising new treatment strategy is way too premature that might generate false hope. It is highly likely that targeting such a central process in the cell will eventually lead to an effective treatment without unacceptable toxic side effects, is not realistic.

This is a very important point. Although we did not observe any evident adverse effects upon treatment with the JNK inhibitor in our animal models (other than reduction in primary tumor growth, lung metastasis and increased sensitization of cancer cells to chemotherapy), further studies are needed to make predictions of side effects in human patients. We are sensitive to the many challenges facing the translation of pre-clinical experimental data into successful clinical treatment. Consequently, we have toned down our statements on the potential therapeutic impact by taking out references to patients and instead suggesting that the work may provide insights for future treatment strategies.

Referee #2 (Comments on Novelty/Model System for Author):

This is a high quality and highly translational study.

Referee #2 (Remarks for Author):

In this manuscript Rodriguez et al describe a series of experiments in which they generate evidence to indicate that metastasis and chemotherapy resistance are JNK-dependent events in breast cancer (basal subtype in particular). They go on to show this is mediated via c-JUN induced transcriptional control of the expression of SPP1 and TNC. This is a solid study and the in vitro and in vivo cell line data are corroborated by frequent reference to patient samples and publicly available datasets. The authors make a strong case for considering the inclusion of JNK inhibitors in standard of care treatment of patients with JNK+++ breast cancers.

There are a few concerns that I would like the authors to consider.

1. It would be helpful to quantitate the data in suppl fig 1A.

We have performed and included a relative protein quantification of p-JNK (Thr183/Tyr185), JNK, p-c-Jun (Ser63) and c-Jun for suppl. fig 1A (now included as Appendix Figure S1A, B and C).

2. There are no stats on Fig 4C & D or Suppl Fig 8D

We have included statistical analyses for the data represented in these figure panels. Details are located in the figure legends.

3. The data in Fig 4D & G (intercepts the Y axis very high) would suggest that SPP1 is not as critical to the metastasis phenotype as is TNC1. The authors may wish to comment on this.

While there may be some differences in the JNK regulation of SPP1 and TNC, it is important to consider also technical variation such as PCR probe efficiency/affinity. Looking at Figure 4D, the results on SPP1 expression is represented with a logarithmic scale shows a 50-60 fold induction in

spheres. TNC expression is induced 45-50-fold. Fig 4G shows the correlation between SPP1/TNC and JUN in metastatic nodules from breast cancer patients. While the overall expression level of SPP1 is higher based on the results, TNC shows a slightly stronger correlation with JUN expression. Similar to the expression analysis using qPCR, here it is also important to consider any technical parameters such as Microarray probe efficiency/affinity. It does look like TNC may be slightly more responsive to JNK inhibition and that TNC knockdown might result in more pronounced reduction in metastasis. However, collectively based on our results, we do not feel confident to make a statement on whether SPP1 is less crucial for the JNK-mediated metastatic phenotype, as compared to TNC. Both genes contribute significantly to metastasis.

4. It was unclear to me whether the data in Fig 5D & E were derived from metastatic foci that had seeded from a primary xenotransplant tumour or whether these were "metastases" that were derived from an iv injection. This should be made clear in the text.

These experiments were performed by intravenous injection and this has been stated more clearly in the text and in the corresponding figure legend.

5. Fig 6 C & D. The reduced metastatic activity is likely explained by the reduced tumour cell numbers following the combination of chemo + SPP1 depletion. This should be stated in the text. Significantly though, it is a very impressive reduction in metastatic activity in the SPP1 deficient cells alone.

It is correct that we cannot exclude that the reduced tumor growth observed under conditions of SPP1 deficiency + chemotherapy (Fig 6B and D) may in part contribute to reduced metastasis. We have adapted the text accordingly. However, when we look individually at the SPP1 deficiency group or the chemotherapy treated group, we see an effect on metastasis without any concomitant changes in primary tumor growth (particularly for SPP1 deficiency, underscoring the importance of SPP1 as a metastasis mediator). This reduction in metastasis is furthered when the two conditions (SPP1 deficiency and chemotherapy) are combined. Observing the data from both mammary tumors and metastases, we feel confident to state that SPP1 deficiency sensitizes the tumors to chemotherapy.

Referee #3 (Remarks for Author):

This carefully executed study confirms and strengthens previous papers on the links between JNK stress signalling, the extracellular matrix (ECM) and aspects of tumorigenesis. Stress initiated by chemotherapies exploits these pathways, hence the inhibition of JNK signalling or disruption of the expression of the emerging ECM proteins SPP1 and TNC sensitizes experimental breast cancer/metastasis models to chemotherapy.

The experiments carried out are technically strong and the link between basal-like breast cancer with significant stem-cell like features and high JNK activity is clearly established. The role of SPP1 and TNC as 'mediators' of JNK signalling is not entirely proven. Although there is a clear link to the c-Jun transcription factor and to tumor progression and chemoresistance no mechanisms are established.

JNK signaling indeed does promote expression of a number of other genes that may be involved in metastasis. However, the evidence that we provide strongly suggests that SPP1 and TNC are directly regulated by JNK signaling and are important mediators of metastasis and therapy resistance. Since targeting ECM proteins is not trivial, one of the important points of our study is that by inhibiting JNK signaling, we can repress two pro-metastatic ECM proteins simultaneously. Nonetheless, we have thoroughly gone through the text to avoid any misunderstanding and adapted when needed. We reveal ECM mediators, downstream of JNK signaling, that promote metastatic progression and chemotherapy resistance. However, we do not provide mechanism of exactly how these ECM proteins promote these effects. TNC is recognized to promote signaling pathways within the metastatic niche and promote cancer cell survival (Oskarsson et al 2011) and we cite this study in our manuscript. We do think that elucidating the mechanistic details of how the ECM proteins mediate chemotherapy resistance is beyond the scope of the current study.

These proteins have previously been implicated in such events and it would be interesting to know if other, perhaps more novel, ECM proteins were evident in the data analyses and may play similar roles.

We agree that it would be highly interesting to analyze other JNK-induced ECM proteins in the context of metastasis and therapy resistance. ECM components such as SPOCK-1, KAL1, S100A16 or PCOLCE2 are regulated by JNK and would surely be interesting as potential candidates. However, their induction in response to active JNK signaling was significantly less when compared to the two matricellular proteins SPP1 and TNC that we focused on. Thus, we believe that, in this initial analysis, the focus on SPP1 and TNC is justified.

The in vitro studies here were confined to tumor cell lines but the role of the stroma in tumorigenesis is acknowledged, but not studied.

In addition to the breast cancer cell lines that we use throughout the study, we have also employed primary cancer cells derived from patient samples in a number of experiments (see for example Fig. 3I and J, Fig. 4C, Fig. EV3C, Appendix Fig. S1 A). However, we do agree with the fact that we have not analyzed the JNK activity that may be induced by cells of the tumor and metastatic stroma (for example in response to chemotherapy). Although we find this a very interesting point, we believe that it is a better fit as the focus of future studies. Nevertheless, we have included comments on this in the discussion.

Some discussion of the relative role of these and other ECM proteins of the tumor and stroma in tumorigenesis and metastasis would be helpful, as well as further acknowledgement of previous work and how this study complements what is already known.

We have expanded the discussion on this point with relevant references.

p.10 line 13 up - the patients were treated with the JNKi

The cancer cells isolated from patient-derived pleural effusion samples were treated in vitro. This has now been stated more clearly in the text.

2nd Editorial Decision

12th July 2018

Thank you for the submission of your revised manuscript to EMBO Molecular Medicine. We have now received the enclosed reports from the referees that were asked to re-assess it. As you will see the reviewers are now globally supportive and I am pleased to inform you that we will be able to accept your manuscript pending the following final amendments:

1) Please address the issues mentioned by referee 1. The Kaplan Meier graph should be removed and the referee's other points should be discussed. We would like to encourage you to replace the graph (Fig. 1A) by Table 1 as we think it would be useful. If you do so, please make sure to update the call outs accordingly.

Please provide a letter INCLUDING my comments and the reviewer's reports and your detailed responses to their comments (as Word file).

REFeree REPORTS.

Referee #1 (Comments on Novelty/Model System for Author):

Some of the issues are well addressed, some seem to be a bit of wind-dressing, some are inadequately addressed.

Referee #1 (Remarks for Author):

Fig 1. Treatment details are lacking. It is not valid to put this wide variety of tumor types, stages, and treatment all in one graph. Such a Kaplan-Meier graph does not provide useful info. From the methodology perspective, this KM-graph is not valid. It should be taken out.

The cited literature regarding pleural effusions from TNBC do not reflect this appropriately. It does not show that pleural effusions are common for TNBC. Indeed, TNBC metastasises to the visceral organs, but this refers to the parenchyma of these organs, rather than the visceral surfaces sec.

The authors do not address the fixation issue sufficiently. It is not true that adipose tissue does not play a role in the quality of fixation.

More importantly, the authors do not provide convincing evidence that JNK activation in the clinical setting is more than just a stress-induced, non-specific phenomenon. They should downplay the impact of their findings, i.e. put their findings in a broader context.

Referee #2 (Comments on Novelty/Model System for Author):

The work presented is of a high technical quality

I believe there is a lot of published data on JNKs and breast cancer behavior. On this basis I would suggest the manuscript is of medium novelty

I believe that the translational impact of this study is some time away. ie I don't believe this will lead to immediate or near term translation or alterations in patient management

Referee #2 (Remarks for Author):

The authors have addressed my concerns adequately

2nd Revision - authors' response

27th July 2018

Point by point responses to editor and reviewer

We thank the editor and reviewer for the insightful comments and have revised our manuscript accordingly. Changes in the manuscript text are highlighted in the red-lined version. Specific points are discussed below.

1) Please address the issues mentioned by referee 1. The Kaplan Meier graph should be removed and the referee's other points should be discussed. We would like to encourage you to replace the graph (Fig. 1A) by Table 1 as we think it would be useful. If you do so, please make sure to update the call outs accordingly. Please provide a letter INCLUDING my comments and the reviewer's reports and your detailed responses to their comments (as Word file).

Reviewer #1 claims that since treatment details are lacking and the analysis includes a number of tumor subtypes and stages, the analysis does not provide useful information. The reviewer is correct in that sample collections with relatively low patient numbers (41 in our case) and some variation in tumor stages and subtypes can be sensitive of confounding effects. To address this, we have analyzed the putative association between JNK activity and a number of parameters, such as “patient age”, “tumor stage” and “receptor status” that are all recognized as established prognostic factors for breast cancer survival. This should address whether JNK activity predicts poor survival with or without association with these parameters. The sample size was appropriate for binomial analysis. The results showed that JNK signaling did not associate with these parameters, yet strongly associated with poor overall survival (in line with the Kaplan Meier analysis). Therefore, a confounding association with any of these parameters is not responsible for the JNK link to poor survival. We have generated a table with this analysis (Appendix Table S2).

With regards to treatment regimen, it is important to consider that: 1) All patients that we analyzed progressed and died from the disease. Thus, it can be assumed that all patients received chemotherapy. 2) The starting and end-points are very well defined.

REFEREE REPORTS

Referee #1 (Remarks for Author):

Fig 1. Treatment details are lacking. It is not valid to put this wide variety of tumor types, stages, and treatment all in one graph. Such a Kaplan-Meier graph does not provide useful info. From the methodology perspective, this KM-graph is not valid. It should be taken out.

Please see the answer to editor – question 1.

The cited literature regarding pleural effusions from TNBC do not reflect this appropriately. It does not show that pleural effusions are common for TNBC. Indeed, TNBC metastasises to the visceral organs, but this refers to the parenchyma of these organs, rather than the visceral surfaces sec.

Our point was not meant to confirm that pleural effusions are particularly common in patients with TNBC, but to provide examples of their occurrence. Importantly, we use both breast cancer cells originating from pleural fluids as well as primary tumors and the results from these cells do line up convincingly.

The authors do not address the fixation issue sufficiently. It is not true that adipose tissue does not play a role in the quality of fixation.

We do not dispute that adipose tissue can affect the quality of tissue fixation. What we can say, is that we did not observe a negative or positive correlation between p-c-Jun staining and amount of adipose tissue in our samples. The adipose tissue was likely not in sufficient quantities to affect the fixation and thus the staining. Importantly, the results from mouse mammary tumors and metastases, where we excluded all adipose tissue, are the same as we observed in the patient samples.

More importantly, the authors do not provide convincing evidence that JNK activation in the clinical setting is more than just a stress-induced, non-specific phenomenon. They should downplay the impact of their findings, i.e. put their findings in a broader context.

It is correct that JNK signaling can be induced by multiple growth factors and stress-stimuli and we do not identify the source of JNK signaling in clinical samples. However, we show that high JNK activity is bad news. Our results indicate that elevated JNK signaling in breast cancer patients, associates with poor response to neoadjuvant therapy and poor overall survival. Identifying what promotes steady state JNK signaling is an interesting question. However, we do not think that this affects the conclusions drawn in our current study and believe it should be addressed in a future study.

Corresponding Author Name: Thordur Oskarsson

Manuscript Number: EMM-2018-09003